



Nitrous oxide emission from pigeon pea – maize rotation in response to conservation
agriculture and biochar amendments in a Ferralsol, northern Uganda
Talent Namatsheve [1,2,*], Vegard Martinsen [1], Jan Mulder [1], Alfred Obia [3], Peter Dörsch [1]
[1] Faculty of Environmental Sciences and Natural Resource Management, Norwegian
University of Life Sciences, P.O Box 5003, 1432 Ås, Norway
[2] Faculty of Plant and Animal Sciences and Technology, Marondera University of Agricultural
Sciences and Technology, P.O. Box 35, Marondera, Zimbabwe
[3] Department of Agronomy, Faculty of Agriculture and Environment, Gulu University, P.O.
Box 166, Gulu, Uganda
[*]Corresponding author: talent.namatsheve@nmbu.no; namatshevetalent@gmail.com
Abstract
Smallholder agriculture in sub-Saharan Africa (SSA) commonly involves limited use of
mineral or organic fertilizer, often resulting in severe nutrient limitation. Conservation
Agriculture (CA), including crop rotation with legumes and biochar amendments, has been
advocated to enhance soil fertility and plant available nitrogen (N). However, CA may affect
nitrous oxide ($N_2O$) emissions even in unfertilized agroecosystems. $N_2O$ is an important
greenhouse gas, and understanding the trade-offs between $N_2O$ emissions and crop yields in
N-poor agroecosystems in SSA is essential. Here we studied crop yield, soil N and $N_2O$
emissions in a double cropping system (pigeon pea – maize rotation) throughout two
consecutive cropping seasons (April-October 2023 and October 2023-January 2024) in a
Ferralsol in Northern Uganda. The study, conducted at a site which had been left fallow for 3
years, involved pairwise comparison of conventionally tilled systems under crop rotation



(Conventional) and continuous maize monocropping (ConventMM). In addition, the effect of
tillage systems (Conventional, CA and CA+biochar) under pigeon pea – maize rotation was
investigated. We defined CA as reduced tillage with planting basins and crop residue retention,
whereas conventional tillage involved overall ploughing. Grain yield was not significantly
affected by rotation or tillage system.  $N_2O$ fluxes were small, ranging from 1.02 – 51.19 µg N
$m^2$ $h^{-1}$ over the entire period. Short-lived emission peaks were observed following pigeon pea
harvest in the crop rotation, which were absent in maize monocropping. Overall, across all
seasons, cumulative growing-season (279 days) $N_2O$ emissions ranged from 0.44 – 1.11 kg N
$ha^{-1}$. Biochar amendments in CA systems did not affect daily $N_2O$ emissions in planting basins.
In the first season, yield-scaled $N_2O$ emissions and N yield scaled $N_2O$ emissions were
significantly smaller in CA systems with biochar compared to conventional tillage, suggesting
that CA and biochar was effective in minimising emissions without penalising pigeon pea
productivity.
Key words: $N_2O$, reduced tillage, legume-cereal rotation, biochar, low input systems.

1.  Introduction
Nitrous oxide ($N_2O$) is a long-lived tropospheric greenhouse gas (GHG) with a lifetime of 116
years, and a global warming potential approximately 300 times greater than that of carbon
dioxide ($CO_2$) (Tian et al., 2020). Atmospheric $N_2O$ is also implicated in the breakdown of
ozone in the stratosphere (Portmann et al., 2012). On a global scale, agriculture is a major
source of atmospheric $N_2O$, contributing approximately 60% to global $N_2O$ emissions
(Adegbeye et al., 2020; Kim et al., 2016). In sub-Saharan Africa (SSA), $N_2O$ emissions are
mainly associated with forest clearing, livestock manure, and crop production (Boateng et al.,
2019; Hickman et al., 2011). Although acidic soils tend to have high $N_2O$ emissions (Wang et





al., 2018), the limited N inputs in smallholder farming systems in SSA reduces soil N
availability, consequently leading to relatively low $N_2O$ emissions. Only ~3% of the globally
applied inorganic fertilizer, a key driver for soil $N_2O$ emissions, is used in Africa (Hickman et
al., 2011). The recent 2024 Nairobi Declaration, targeting increased fertilizer use in Africa
(Africa Union, 2024), might change future trajectories of fertilizer consumption in SSA,
potentially increasing soil N availability and $N_2O$ emissions over time. Besides increasing
mineral N fertilization, additional plant-available N can be derived from introducing legumes
in crop rotation (Jensen et al., 2012), or using organic fertilizers such as animal manure. These
strategies are central to conservation agriculture (CA) in subsistence farming systems, but little
is known about how they affect baseline $N_2O$ emissions. Also, climate smart practices such as
biochar amendments, have been proposed to enhance crop yield and soil fertility (Namatsheve
et al., 2024; Schmidt et al., 2021) and to reduce $N_2O$ emissions (Zhang et al., 2021).
Nitrous oxide is an intermediate or by-product in soil N transformations, that include
nitrification, denitrification and nitrifier denitrification (Meier et al., 2020). The
biogeochemistry of $N_2O$ in soil is to a large extent regulated by complex interactions between
environmental and biological factors such as temperature, water, oxygen levels, acidity and
substrate availability (Case et al., 2015; Tian et al., 2020). Nitrification occurs under
predominately aerobic soil conditions, whereby autotrophic bacteria or archaea oxidize $NH_4^+$
to $NO_2^-$, which is further oxidized to nitrite ($NO_3^-$) by nitrate-oxidizing bacteria (Dick et al.,
2008; Fungo et al., 2019). Denitrification occurs in predominately anaerobic soils, or soil
aggregates, and is an anoxic respiratory process mediated by bacteria and some fungi, reducing
$NO_3^-$ stepwise to $N_2$ via the intermediates $NO_2^-$, NO and $N_2O$ (Saggar et al., 2013; Scheer et
al., 2020). Nitrifier denitrification occurs when nitrifying bacteria reduce $NO_2^-$ under hypoxic
conditions, analogously to the denitrification pathway (Wrage-Mönnig et al., 2018).



In a quest to improve crop production and soil fertility, sustainable agricultural practices such
as conservation agriculture (CA) and biochar amendment have been promoted in SSA
(Namatsheve et al., 2024). Conservation agriculture may improve crop production (Giller et
al., 2015; Hobbs et al., 2008) and is based on three core principles: the first principle is
minimum soil disturbance, which may enhance water retention and soil organic matter content
(Pittelkow et al., 2015; Powlson et al., 2011). In addition to increased crop production, this may
lead to increased $N_2O$ emission (Guenet et al., 2021; Shakoor et al., 2021). Availability of N in
non-fertilized systems can be improved through the second principle of CA, which involves
incorporating legumes in cereal dominated farming systems (Namatsheve et al., 2021) and the
third principle, crop residue retention (Fang et al., 2007; Turmel et al., 2015), both of which
may stimulate $N_2O$ emissions (Abalos et al., 2022). Mitigation of $N_2O$ emission in low-input
crop production systems will ultimately depend on synchronizing the release of mineral N from
legumes and crop residues with N uptake by crops. Besides CA, also, biochar, a carbon rich
material produced by pyrolysis of organic waste (Cornelissen et al., 2016; Lehmann, 2007),
has been claimed to enhance crop production, but its role on N cycling in unfertilised systems
remains unclear. In addition, biochar tends to increase soil pH which favours $N_2$ over $N_2O$ as a
main product of denitrification (Obia et al., 2015; Wang et al., 2018). Although biochar
contributes to the retention of exchangeable plant-available $NH_4^+$, it also may immobilize soil
N (Nguyen et al., 2017), thereby reducing N availability and $N_2O$ emission (Jeffery et al., 2015;
Namatsheve et al., 2024). However, Weldon et al., (2022) reported that the magnitude for
sorption capacity of biochar for $NH_4^+$ is low, with a huge uncertainty range.
Indeed, earlier studies reported increased $N_2O$ emissions in SSA under CA (Baggs et al., 2006;
Raji and Dörsch, 2020; Shumba et al., 2023), while biochar amendments have been shown to
reduce $N_2O$ emissions (Fungo et al., 2017, 2019; Namoi et al., 2019). However, these studies
were carried out in systems that received inorganic N fertilizers, which do not represent the





realities of unfertilized smallholder tropical agroecosystems typical of Uganda and other
countries in SSA. Our meta-analysis indicated that residue retention increases $NO_3$ availability
and subsequently $N_2O$ emissions (Namatsheve et al., 2024). Building on this, we recently
demonstrated that integrating biochar into CA systems enhanced biological $N_2$-fixation of
pigeon pea in unfertilized systems of Uganda (Namatsheve et al., 2025). This raises questions
on the implication of the additional N from biological $N_2$-fixation on $NO_3$ and $N_2O$ emissions
in unfertilized systems, with tight N cycling. As far as we know, there are no published studies
that examine the synergy of CA and biochar on $N_2O$ emission in unfertilized tropical
agroecosystems.
In this study we investigated the effect of conservation agriculture on grain yield, $N_2O$
emissions, mineral N dynamics, and yield-scaled $N_2O$ emissions on an unfertilized Ferralsol in
northern Uganda over two consecutive cropping seasons. Specifically, we compared crop
rotation (pigeon pea – maize) with maize monocropping under conventional tillage (overall
digging). In addition, we compared pigeon pea – maize rotation under three practices, i.e.
conventional tillage, CA (reduced tillage), and CA in combination with biochar (CA+biochar).
We hypothesised that rotation with pigeon pea increases $N_2O$ emission compared to maize
monocropping, while CA+Biochar reduces $N_2O$ emissions, both compared to CA and to
conventionally tilled soil.
2. Methodology
2.1. Site description
A field experiment was carried out in Gulu, Northern Uganda (*2° 47′ 46″ N, 32° 20′ 45″ E*).
Uganda has a bimodal rainfall pattern and eight distinct agro-ecological regions, where Gulu
lies in the Northern savannah grasslands (Mubiru et al., 2012). Soils in Gulu are acric
Ferralsols, and the texture is a loamy sand (Wortmann and Eledu, 1999). Average soil organic



C and total N are 1.52% w/w and 0.11% w/w, respectively, while average soil pH is 6.71 (Table
S19). The research site has a double cropping system i.e., one during the first rain season from
April to August and a second one during the second rain season August to December, a dry
period is from December to February. The average annual temperature is 24 °C, and the annual
rainfall in 2023 was 1238 mm, of which 818 mm was received in the first season and 419 mm
in the second. The weather data were obtained from the Gulu weather station which is about 6
km from the experimental site.
2.2. Experimental design, crop establishment and management
The experiment was established on a field that has been a fallow for the 3 previous years; before
that it was used for maize and cassava production without mineral fertilizer. Prior to the
establishment of the experiment, a dense vegetation of grasses was removed by slashing and
chemical weeding using glyphosate [N-(phosphonomethyl) glycine]. Plots (6 m × 5 m) under
conventional management were prepared by overall digging using hand hoeing (100% tillage)
and plots of the same size under CA by manually digging 10-L planting basins (35cm long ×
15cm wide × 20cm deep) spaced 70 cm × 35 cm (interrow × within row spacing). Given these
basin dimensions about 10% - 12% of the land under CA was tilled. The experiment had four
treatments, replicated four times, and randomised in a complete block design (RCBD). The
treatments were ConventMM, Conventional, CA and CA+BC (Table 1). Biochar was
homogeneously mixed into the basins of the CA+BC plots when preparing the planting basins
before sowing. In the treatments with crop rotation, pigeon pea was sown in the 1st season, and
maize in the 2nd season. Maize monocropping had maize in both seasons. Dates of sowing and
harvesting are indicated in Fig. 1a.
A pigeon pea variety SEPI 1 (bred at ICRISAT, Malawi and released by the National
Agricultural Research Organisation, Uganda) was sown uniformly in Conventional, CA and



CA+BC treatments. SEPI 1 is a medium maturity variety, with 77 – 87 days to flowering and
105 – 139 days to 75% maturity. It is an indeterminate variety with semi-branching growth, the
main stem continuing to elongate indefinitely; potential grain yields range from 1.8 – 3.4 Mg
ha$^{-1}$. A maize variety, Longe 10H, which is a hybrid variety with 100 – 120 days to maturity
and a yield potential of 7 – 9 Mg ha$^{-1}$ was sown in the ConventMM treatment (1$^{st}$ season) and
in all treatments in the 2$^{nd}$ season. During sowing in CA treatments, three seeds were planted
in each basin spaced 10 cm from each other giving a total planting population of 56 000 plants
per ha, equalling planting population in conventional tillage treatments.
To mimic subsistence farming systems in Uganda, no inorganic fertilizer was applied. For CA,
weeds were controlled by spraying glyphosate at a rate of 1.03 L ha$^{-1}$, immediately after sowing
and hand pulling throughout the season. Weed control in the conventional treatment was done
by hand hoeing at planting and throughout the season.
2.3. Biochar production and application
Biochar was prepared from pigeon pea stems and twigs using the flame curtain "Kon Tiki" kiln
(Cornelissen et al., 2016; Munera-Echeverri et al., 2020). The kiln consists of a conically
shaped pit with a depth of 1m and a diameter of 3 m. The pyrolysis temperature was 600 °C.
After weighing, the pigeon pea feedstock was pyrolyzed, quenched with water, covered with
banana leaves and soil, and recovered after 3 days. The biochar was weighed (dry matter),
ground and packed. The feedstock to biochar conversion ratio was 4:1, and the biochar had a
pH of 9.74, carbon (C) concentration of 51%, nitrogen (N) concentration of 0.76%, cation
exchange capacity (CEC) of 80.94 cmol$_c$ kg$^{-1}$ and plant available P of 703 mg kg$^{-1}$ (Table S8).
During biochar application, manually dug 10-L planting basins (35cm long × 15cm wide ×
20cm deep) spaced 70 cm × 35 cm (interrow × within row spacing) were opened and the soil
was mixed evenly with 1 litre of biochar (240g w/w) per basin for the CA+BC treatment, and



500 ml of biochar (120g w/w) for the CA+BC+BC treatment in the first year, and the other
120g w/w in the second year. Afterwards, the planting basins were covered with a thin layer of
soil.

Table 1: Treatment description and management of the experiment site at Gulu, Northern
Uganda

| Treatment name and abbreviation | Treatment description and management |
|---|---|
| 1. Conventional tillage, maize monocropping, no rotation (ConventMM) | Shallow conventional tillage by hand hoeing (overall digging) to a depth of 10 cm (100% tillage). Maize was grown in both seasons. Crop residues were left on the soil surface and spread evenly after harvesting. Plant spacing 70 cm × 35 cm (interrow × within row spacing). Weed control by hand weeding during sowing and throughout the season. |
| 2. Conventional tillage, pigeon pea – maize rotation (Conventional) | Shallow conventional tillage by hand hoeing (overall digging) to a depth of 10cm (100% tillage). Pigeon pea – maize rotation; crop residues were left on the soil surface and spread evenly after harvesting, while stems were removed from the field. Plant spacing 70 cm × 35 cm (interrow × within row spacing). Weed control by hand weeding during sowing and throughout the season. |
| 3. Basins, pigeon pea – maize rotation, incorporating residues (CA) | Manually dug 10-L sized planting basins (approximately 35 cm long, 15 cm in diameter and cm deep) created by hand hoeing at the beginning of the experiment with a spacing of 70 cm × 35 cm (interrow × within row spacing). Pigeon pea – maize rotation; crop residues were left on soil surface and spread evenly after harvesting. Weed control by herbicides (glyphosate) during sowing and hand pulling throughout the season. |
| 4. Basins, pigeon pea – maize rotation, incorporating residues, 4 Mg ha$^{-1}$ of biochar applied once (CA+BC) | As CA, but with biochar mixed into the basins at a rate of 4 Mg ha$^{-1}$ during the first season, before sowing. |




2.4. Soil sampling and analysis for chemical characterization
Soils were sampled before establishing the trials in March 2023 (background sampling) for
general characterisation of the research site. Using an auger, 3 samples from 0-20 cm depth
were randomly taken from the experimental site and mixed into a single composite sample
(Table S19). Plot wise sampling was carried out at the onset (April 2023) and end (October
2023) of the first growing season from planting basins in CA and CA+Biochar treatments and
in the planting rows in conventional treatments to assess the effect of different treatments on
soil properties. The soils were sampled at $0 - 20$ cm depth in each plot. Prior to analysis,
samples from each treatment in each of the four blocks were bulked (viz., n=4 for the onset and
n=4 for the end for each treatment), air dried and passed through a 2 mm sieve. Soil pH was
determined in water (2.5:1) (Gee and Bauder, 1986). Soil organic carbon (SOC) and N was
determined using a Thermo Finnigan EA attached to the Elemental Analyzer Analysis-Isotope
Ratio Mass Spectrometry (EA-IRMS).

2.5. Nitrous oxide flux sampling and analysis
The static chamber method was used to estimate $N_2O$ emissions. We used cylindrical 20 cm
high, custom-made PVC chambers manufactured from 16 cm diameter, grey opaque sewage
pipes with a self-sealing rubber septum on the top for gas sampling. Permanent gas sampling
plots were established by inserting 17 cm diameter PVC rings (the base) to a depth of 7 cm into
the soil in April 2023, during sowing in the 1st season. We used two chamber positions in each
plot. One was placed in the interrow, between two rows and another in-row between two plants
within a row or in the planting basin (CA) (Fig S1, S2).
The chambers were deployed by carefully inserting them 3 cm into the pre-installed collars to
obtain an airtight fit. To facilitate chamber deployment, the contact area between the collar and





chamber was sealed with a thin layer of petroleum jelly. Each chamber covered an area of 0.020
$m^2$ and had a total headspace volume of 0.004 $m^3$. For each flux measurement, four gas samples
were drawn from the chamber headspace 1, 15, 30 and 60-minutes after deployment using a 20
mL polypropylene syringe equipped with a three-way valve. Collected gas samples were
transferred to pre-evacuated 12 mL glass vials, which were crimp-sealed with butyl septa.
When sampling the chamber headspace, the plunger of the syringe was moved slowly in and
out for three times to mix the gas and obtain a representative sample. Air and chamber
temperature was recorded before removing the chambers using a handhold thermometer which
was placed inside a chamber, before and after sampling.
Gas sampling was done approximately biweekly, resulting in 17 sampling campaigns between
May 2023 and January 2024. The vials were shipped to the Norwegian University of Life
Sciences for $CO_2$ and $N_2O$ analysis by gas chromatography. $CO_2$ measurements were done for
quality check. He-filled vials were included as blanks to check for contamination during
storage and shipment of the vials. The vials were analysed on a gas chromatograph (GC; model
7890A, Agilent, Santa Clara, CA, USA) connected to an auto-sampler (Gilson). $N_2O$ was
quantified by an electron capture detector and $CO_2$ by a thermal conductivity detector as
described by (Žurovec et al., 2017).
Flux rates were estimated by fitting a linear or second order (polynomial) function to the
concentration change over time in the chamber headspace. Changes in concentration were
converted to area flux as follows:

$$F = \frac{dc}{dt} \cdot \frac{Vc}{A} \cdot \frac{Mn}{Vn} \cdot 60,$$


where F is the $N_2O$ flux ($\mu$g $N_2O$-N m$^{-2}$ h$^{-1}$), $\frac{dc}{dt}$ the rate of change in concentration over time
in the chamber headspace (ppm min$^{-1}$), Vc the volume of the chamber, 0.004 $m^3$, A the area





covered by the chamber, 0.020 m$^2$, Mn the molar mass of N in N$_2$O (g mol$^{-1}$) and Vn the
molecular volume of N$_2$O or CO$_2$ at chamber temperature (m$^3$ mol$^{-1}$). A quadratic fit was used
if N$_2$O concentration in the chamber showed a convex downwards trend with time. Fluxes were
cumulated plot-wise by linear interpolation. The cumulative N$_2$O emissions (kg N$_2$O-N ha$^{-1}$)
were calculated as follows:
$$cumulative\ N_2O\ =\ \sum (f_i\ +\ (f_{i+1})/2 \times (t_{i+1} - t_i) \times 24\ \times 10^{-5}$$
where $f$ represents the N$_2$O flux ($\mu$g N$_2$O-N m$^{-2}$ h$^{-1}$), $i$ the $i$th measurement, $(t_{i+1} - t_i)$ the
number of days between two adjacent measurements, and $24 \times 10^{-5}$ was used for unit
conversion. We also scale up the N$_2$O-N cumulative emissions to hectare using a scaling factor
of 0.12 for basin and 0.88 for interrows in CA treatments, while 0.50 was used for both inrows
and interrows in conventional treatments.
2.6. Soil moisture and mineral N content
Directly after each flux sampling, soils were sampled from both planting stations (Conventional
and ConventMM) and basins (CA and CA+Biochar) and from interrow positions (all
treatments), for analyzing mineral N (NO$_3$-N and NH$_4^+$-N) and soil moisture. Soils were
sampled from 0-20 cm depth, using a 10 mm diameter corer with a height of 20 cm. Only one
core was taken to prevent excessive perturbation, particularly in the planting basins. The cores
were stored in a cooling box on ice and shipped to Gulu University which is located 5 km from
the experimental site. Mineral N was extracted from 11g of field moist soil in 40 ml of 2M
potassium chloride (KCl), after 1 hour of horizontal shaking at 200 strokes per minute and
passing the supernatant through Whatman filters grade 589/3. The supernatants were frozen for
subsequent analysis of NO$_3$-N and NH$_4^+$-N at the Norwegian University of Life Sciences by
flow injection analysis (FIA star 5020, Tecator, Sweden).



The remaining soil was dried at 105°C for 72h to determine gravimetric moisture content and
bulk density (BD). BD was calculated by dividing weight of oven dried soil with the volume
of the soil core (15.714 cm$^3$) and its gravimetric soil moisture content calculated by dividing
weight of water (difference between fresh soil weight and oven dried soil) by the weight of
oven dried soil.
The bulk density (BD) was then used to calculate water filled pore space (WFPS) as follows:
$$WFPS = \frac{\theta g \times BD}{\left(1 - \frac{BD}{PD}\right)} \times 100$$

where $\theta g$ is the gravimetric water content, *BD* the soil bulk density (1.29 ± 0.01 g cm$^{-3}$) and
*PD* the soil particle density (2.65g cm$^{-3}$). Daily rainfall and temperature data were obtained
from the Gulu meteorological station which is located 6 km from the experimental site.
2.7. Yield and yield-scaled emissions
Crops were harvested at physiological maturity, 6 months after sowing for pigeon pea and 4
months for maize. To compare crop yields under conventional and CA management and in
CA+BC treatments, all values for dry biomass and grain (moisture content of 12.5% for maize
and 15% for pigeon pea) were extrapolated from the plot to the hectare. Yield-scaled $N_2O$
emissions (kg $N_2$O-N kg$^{-1}$ grain yield) and N yield scaled emissions (kg $N_2$O-N kg$^{-1}$ grain N)
were estimated for each season by dividing the scaled cumulative $N_2O$ emissions with grain
yield or N content of the grain (N concentration x grain yield). A scaling factor of 0.12 for $N_2O$
emissions in planting basins and 0.88 for $N_2O$ emissions in interrows was used to calculate
yield-scaled $N_2O$ emissions in CA and CA+BC treatments. In conventional treatments, scaling
factor of 0.50 was used for both inrows and interrows.



2.8. Data analysis
All data were analysed using R software, version 4.3.2. A random intercept, fixed slope linear
mixed-effect model using the *lmer* function from *lme4* packages (Bates, 2010) with treatment,
chamber position (interrow and in row) and season as fixed factors was used to evaluate
treatment effects on $N_2O$ emissions and soil mineral N. On soil parameters, yield and yield-
scaled emissions, fixed factors were treatments and seasons while block was a random factor
(Table S1 – S20). Variation associated with sampling day (17 levels) and blocks was modelled
by introducing random effects to account for repeated measurement on the same plot, on $N_2O$
fluxes and soil mineral N. The most parsimonious model was selected after model comparisons
based on goodness of-fit assessed by the Akaike Information Criterion (AIC) and the Bayesian
Information Criterion (BIC), and stepwise model reduction (Aho et al., 2014). We assumed
normal error structure and homoscedasticity and validated the model assumptions by checking
quantile plots of residuals against fitted values (Zuur et al., 2009). Model parameters (estimated
marginal means) were extracted using the "*emmeans*" package (Lenth, 2016), and multiple
comparisons were performed using multcomp (Hothorn et al., 2008) with adjusted *p* values
(Tukey post-hoc test at 0.05 probability level) (Lenth, 2016). The 95% confidence intervals
(CI) were retrieved using lsmeans function. Differences between levels of the fixed effects
were assessed using multicomp package. Linear regression analyses were performed to analyse
the relationship between $N_2O$ fluxes with WFPS and mineral N. Visualization of the fitted
models was achieved using the package ggplot2 (Wickham, 2016).






3.0 Results
3.1. Soil parameters
Soils were near neutral with a background pH of 6.71 (Table S18). During the onset of the first
season, soil pH ranged from 6.71 – 6.97. CA+BC significantly ($p < 0.05$) increased pH
compared to CA systems at the onset of the first season (Table 2). Generally, pH decreased
from the beginning to the end of the first season, after which no significant pH differences
among treatments were found. SOC ranged from 1.25 – 2.23% and biochar significantly
increased C, from 1.30% in CA to 2.23% in the CA+BC treatment (Table 2). At the end of the
first season, CA+BC had significantly higher C than other non-BC treatments. Different
treatments did not affect soil mineral N at the beginning and end of the first season (Table 2).



Table 2: Treatment effects on inrow (conventional treatments) and within planting basin (CA
treatments) soil properties (pH, C, N, and BD). Soils were sampled at sowing (onset) and at
the end of the first season, in Gulu, Uganda. Means are shown with standard errors of means.

| | pH | | C (%) | | N (%) | |
|---|---|---|---|---|---|---|
| Treatment | onset | end | onset | end | onset | end |
| Conventional | 6.64±0.14 b A | 6.36±0.10 a A | 1.41±0.06 b A | 1.31±0.07 b A | 0.11±0.01 a A | 0.10±0.00 a A |
| CA | 6.61±0.08 b A | 6.35±0.07 a A | 1.30±0.04 b A | 1.25±0.05 b A | 0.10±0.01 a A | 0.09±0.00 a A |
| CA+BC | 6.97±0.13 a A | 6.28±0.20 a B | 2.23±0.27 a A | 2.15±0.11 a A | 0.11±0.01 a A | 0.12±0.00 a A |
| ConventMM | 6.71±0.04 ab A | 6.59±0.12 a A | 1.52±0.18 b A | 1.35±0.06 b A | 0.12±0.01 a A | 0.10±0.00 a A |
| *P-value* | <0.001 | 0.364 | <0.001 | *<0.001* | 0.569 | *0.569* |


Lowercase letters compare treatments at onset or end of sampling, while uppercase letters
compare change between samples taken at onset and end of the season. Different letters
represent significant differences (p < 0.05), determined at 5% level using Tukey test.



3.2. $N_2O$ emission dynamics

Treatment and chamber position significantly ($p<0.05$) affected $N_2O$ hourly fluxes (Table S2, S4). $N_2O$ fluxes were small in all treatments throughout the entire observation period (Fig. 1a) with treatment averages ranging from $1.02 – 51.19$ µg m$^{-2}$ h$^{-1}$. The fluxes peaked in mid-October following pigeon pea harvest and sowing of maize, were conventionally tilled soil with pigeon pea – maize rotation (Conventional) had the largest emissions, while conventional tillage with maize monocropping (ConventMM) showed a far less pronounced $N_2O$ emission peak in October. $N_2O$ emissions levelled off towards the end of the second season, in January, coinciding with a longer dry spell (Fig 1a).

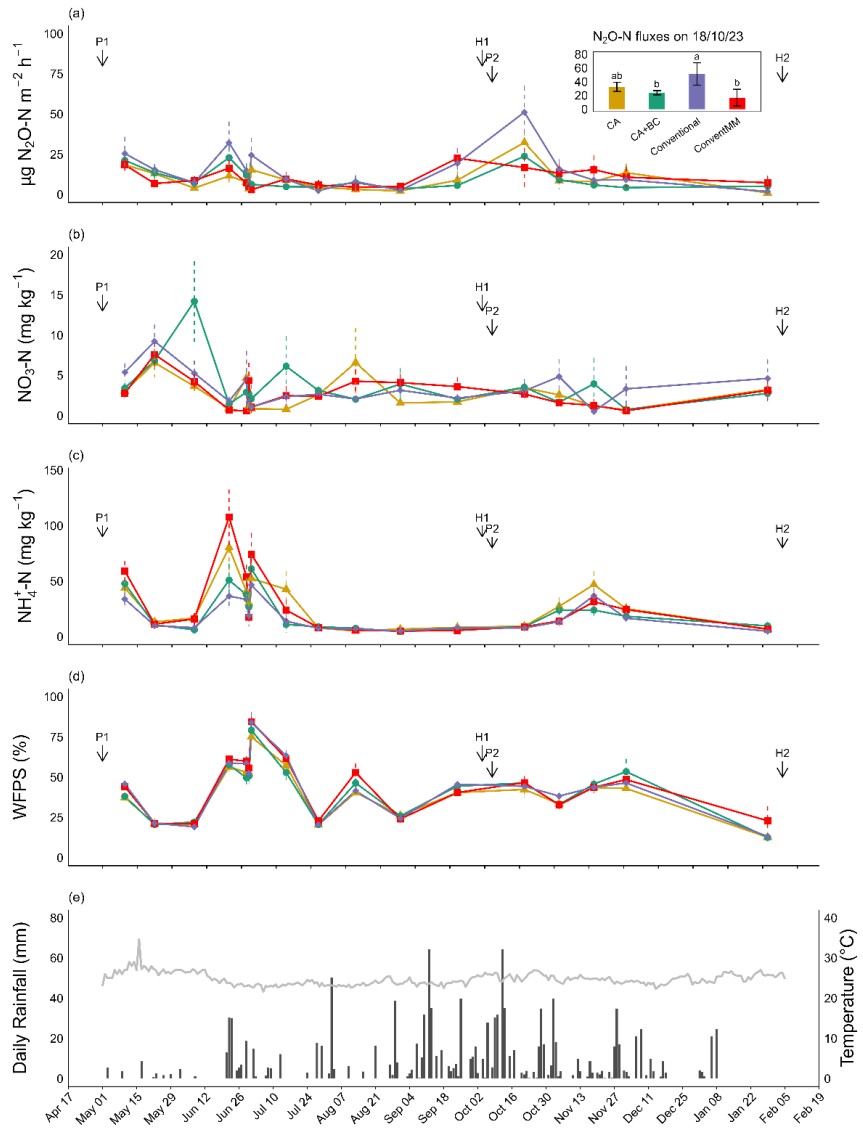

Fig 1: Mean (± se) of (a) $N_2O$ emission fluxes, (b) KCl-extractable $NO_3^-$ and (c) $NH_4^+$, (d) WFPS, as well as (e) daily rainfall and air temperature (℃) during the two cropping seasons between May 2023 – January 2024 in Gulu, Uganda. P1 and P2 indicate planting date for the 1st and 2nd second season (18 April 2023 and 05 October 2023, respectively). The data is based on 8 observations, average of 2 chamber positions and 4 blocks. H1 and H2 indicate harvesting dates (01 October 2023 and 30 January 2024, respectively. The insert in (a) shows mean ± se $N_2O$ fluxes during peak emission on 18 October 2023.



### 3.3. Mineral N dynamics

$NO_3^-$N concentrations ranged from $0 – 15$ and $0 – 5$ mg kg$^{-1}$ in the first and second season, respectively (Fig 1b), while $NH_4^+$ ranged from $10 – 110$ and $10 – 45$ mg kg$^{-1}$ in the first and second respectively (Fig 1c). Generally, both $NH_4^+$ and $NO_3^-$ were more variable in season one than in season two (Fig 2). The rotation effect on $NH_4^+$ was significant in the first season ($p <$ 0.05), with more extractable $NH_4^+$ in the ConventMM (31.22 mg kg$^{-1}$, $20 – 42.40$ CI) than the Conventional treatment (19.20 mg kg$^{-1}$, $8 – 30.50$ CI). (Fig 2a).

Tillage system significantly ($p < 0.05$) affected $NH_4^+$, CA (27.19 mg kg$^{-1}$, $22.5 – 31.9$ CI) increased $NH_4^+$ compared to other treatments. $NO_3^-$ ranged from $0 – 14$ mg kg$^{-1}$ and there was no significant ($p > 0.05$) season and treatment effect (Fig 3b). $N_2O$ fluxes were affected by Mineral N ($NH_4^+$ and $NO_3^-$), but with a weak coefficient of determination ($R^2 = 0.04$, $p < 0.001$) (Fig 4).



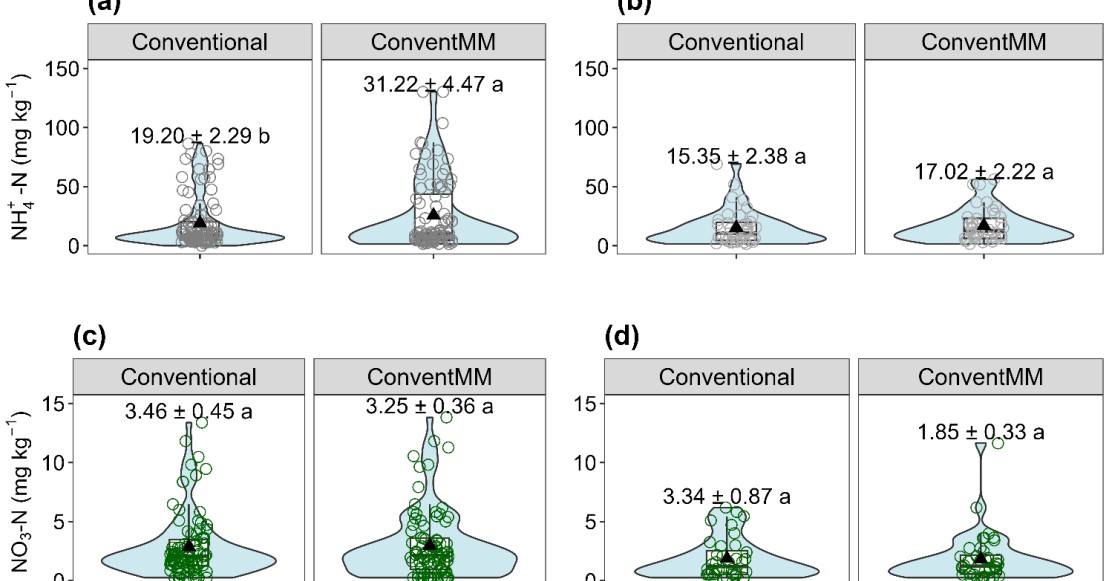

Fig 2: Violin and box-whisker plot showing the effect of rotation on $NH_4^+$ in season 1 (a) and season 2 (b), and $NO_3^-$ in season 1 (c) and season 2 (d) in Gulu, Uganda. Upper and lower edges of boxes indicate 75th and 25th percentiles, horizontal lines within boxes indicate median, whiskers below and above the boxes indicate the 10th and 90th percentiles, and triangles indicate arithmetic mean. Differences between treatments were tested using the Tukey post-hoc test. Different letters represent significant differences ($p < 0.05$).



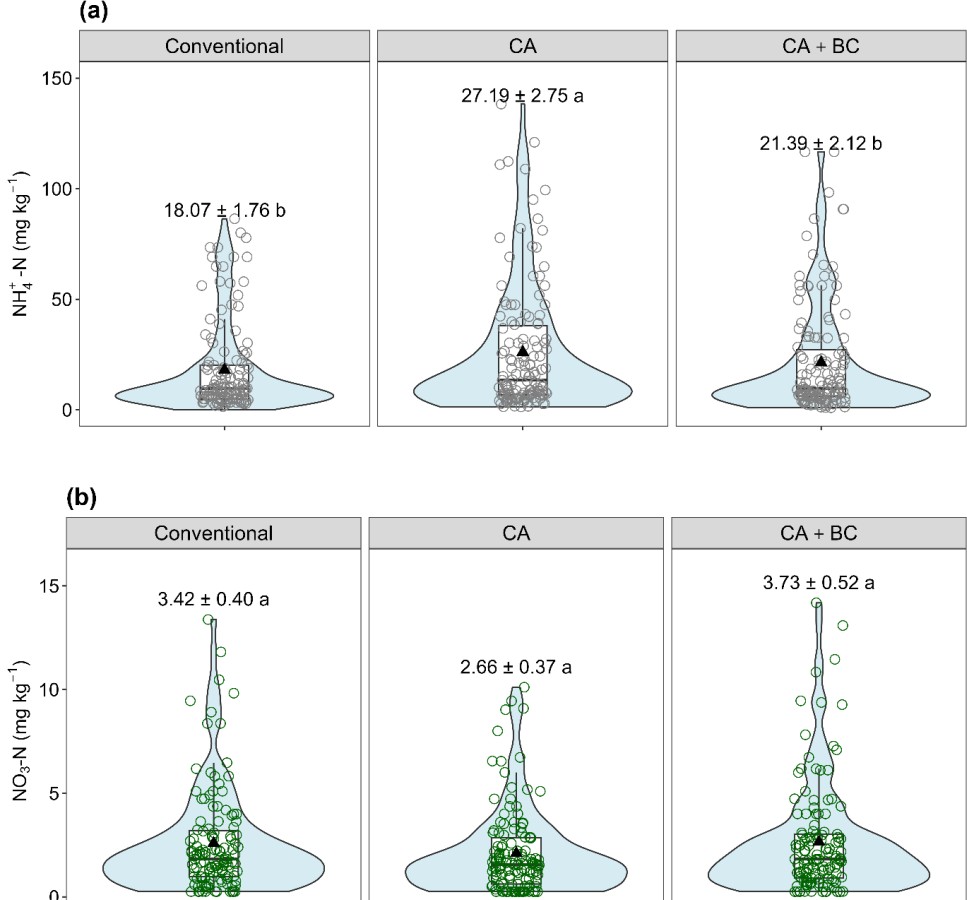

Fig 3: Violin and box-whisker plot showing the effect of tillage systems on (a) $NH_4^+$ and (b) $NO_3^-$ for both seasons in Gulu, Uganda. Upper and lower edges of boxes indicate 75th and 25th percentiles, horizontal lines within boxes indicate median, whiskers below and above the boxes indicate the 10th and 90th percentiles, and triangles indicate arithmetic mean. Data plotted is the average of season 1 and 2. Significant differences (p <0.05) between treatments were tested using the Tukey post-hoc test. Different lowercase letters indicate differences between treatments within a season.



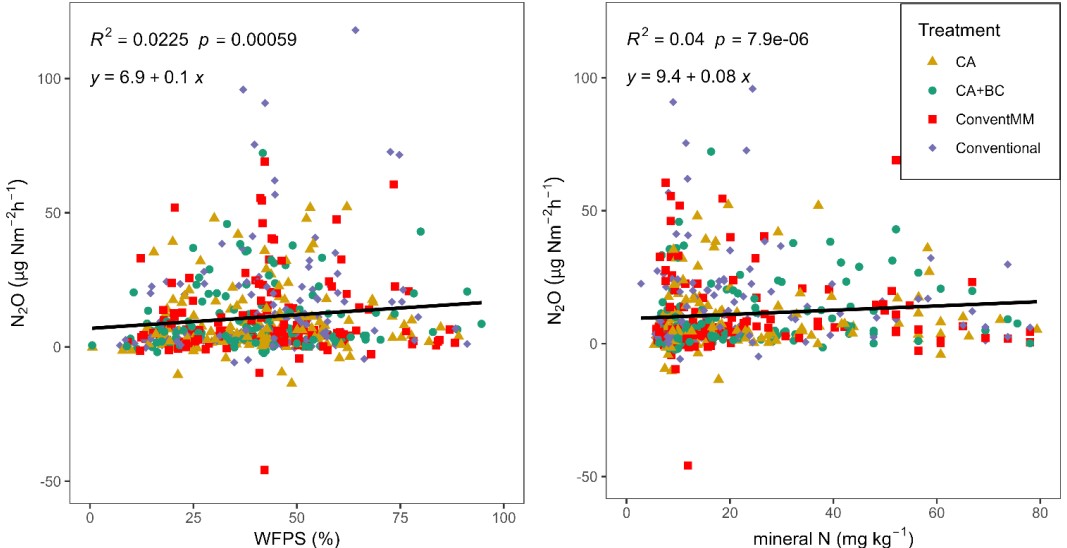


Fig 4: Relationship between hourly $N_2O$ fluxes and water filled pore space (WFPS) and mineral

N (mg kg$^{-1}$) in Gulu.  Data points are for 17 sampling dates.


3.4. Effect of rotations and tillage on cumulative $N_2O$ emissions
Cumulative $N_2O$ emissions ranged from 0.24 – 0.50 kg N ha$^{-1}$ in the 1st season (134 days), and
from 0.19 – 0.61 kg N ha$^{-1}$ in the 2nd season (125 days). Cumulative $N_2O$ emissions for the
entire measurement period May 2023 to January 2024 ranged from 0.44 – 1.11 kg N ha$^{-1}$ (279
days) (Fig. 4, Fig. 5, Fig. S3). Chamber position did not significantly explain variations in
cumulative $N_2O$ emissions and there were no significant differences (p > 0.05) between
conventionally tilled pigeon pea – maize rotation and conventionally tilled continuous maize
monocropping (Table S17, S18, Fig. 5, Fig S3). Treatments and the interaction of treatment
and position showed a significant effect (p < 0.05) of tillage system (Conventional, CA, and
CA+BC) on cumulative $N_2O$ emissions in season 1 and for the entire sampling period (season
1 and 2 combined) (Table S20). Cumulative $N_2O$ emissions in season 2 were only affected (p





$< 0.05$) by treatment (Fig. 6). There was no significant difference in cumulative $N_2O$ between
season one and two (Fig. 6a, b). Average cumulative $N_2O$ were highest in conventional interrow
with no significant differences between the other treatments, irrespectively of position. The
cumulative $N_2O$ for the whole sampling period (season 1 and 2) was significantly smaller under
CA+BC compared to conventional in the interrows (Fig 6c). Significantly ($p < 0.05$) larger
emissions were recorded in inter-rows compared to in-rows in conventional treatment during
the first season (Fig 6).

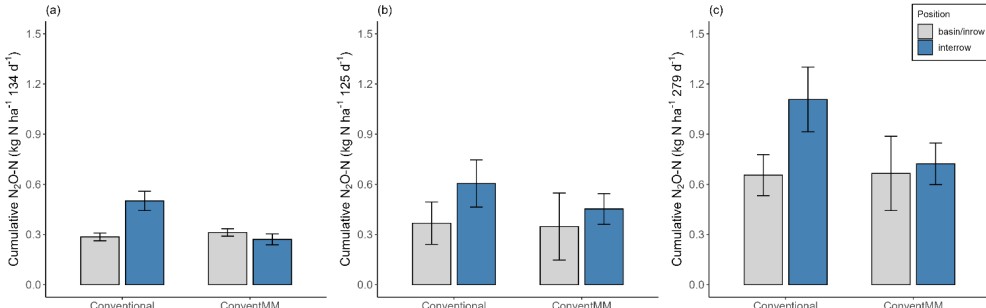


Fig 5:   Cumulative $N_2O$ emissions (kg N ha$^{-1}$) under conventional tillage with pigeon pea
rotation (Conventional) and maize monocropping (ConventMM) in Gulu, Uganda. Shown are
treatment arithmetic means and SE for conventionally tilled continuous maize monocropping
(MM) and conventionally tilled PP-maize rotation in (a) first season (May 2023 to September
2023), (b) second season (October 2023 to January 2024) and (c) sum of both seasons (May
2023 – January 2024).



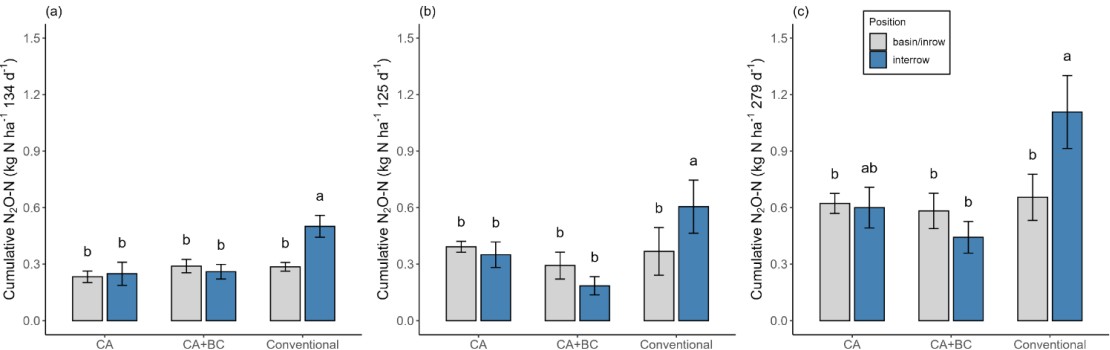

Fig 6: Cumulative N$_2$O emissions (kg N ha$^{-1}$) for systems under pigeon pea – maize rotation

(a) first season with pigeon pea (measured from May 2023 to September 2023), (b) second

season with maize (measured from October 2023 to January 2024) and (c) sum of both seasons

(May 2023 – January 2024), in Gulu, Uganda. Error bars represent standard errors (N = 4).

Different letters represent significant differences between treatments for each of the seasons or

the cumulative of all the seasons (p <0.05) using the Tukey post-hoc test.





Table 3: Grain yield, grain N yield, yield scaled $N_2O$ emissions and N yield scaled $N_2O$
emissions during the first and second rain season, in northern Uganda. Means are shown with
standard errors of means.

| Season | Treatment | Crop | Grain yield (kg ha$^{-1}$) | Grain N yield | Yield scaled emissions (g $N_2O$-N kg$^{-1}$ grain) | N yield scaled emissions (g $N_2O$-N kg$^{-1}$ N grain) |
|---|---|---|---|---|---|---|
| First season | ConventMM | Maize | 856.67±157.12 B | 10.46±2.17 B | 0.36±0.04 A | 29.86±3.44 A |
| | Conventional | pigeon pea | 1266.67±83.78 a | 40.30±2.75 a | 0.32±0.04 a | 9.93±1.05 a |
| | CA | pigeon pea | 1539.17±175.70 a | 50.79±6.18 a | 0.16±0.03 b | 4.95±0.85 b |
| | CA+BC | pigeon pea | 1580.50±89.67 a | 50.73±2.76 a | 0.18±0.03 b | 5.31±0.97 b |
| | *P-value* | | ns | ns | < 0.05 | < 0.05 |
| Second season | ConventMM | maize | 2590.91±156.72 a A | 29.86±2.17 a A | 0.15±0.04 a B | 13.46±3.87 a B |
| | Conventional | maize | 2658.20±122.73 a | 34.82±4.08 a | 0.19±0.05 a | 14.83±4.13 a |
| | CA | maize | 2743.51±310.59 a | 34.42±3.23 a | 0.14±0.01 a | 10.39±1.65 a |
| | CA+BC | maize | 2758.31±175.24 a | 39.27±2.76 a | 0.08±0.02 a | 5.11±1.34 a |
| | *P-value* | | ns | ns | ns | ns |








3.5.     Treatment effects on grain yield and yield scaled $N_2O$-N emissions
Different treatments did not affect ($p > 0.05$) grain yield in neither season. Pigeon pea grain
yield ranged from $1.3 - 1.6$ t ha$^{-1}$ in the first season, and maize grain yield ranged from $2.6 -$
$2.8$ t ha$^{-1}$ in the second season. Grain N yield was also not affected by treatments and it ranged
from $10 - 51$ kg N ha$^{-1}$ in the first season, and $30 - 39$ kg N ha$^{-1}$ in the second season (Table
3). Yield scaled emissions and N yield scaled emissions ranged from $0.16 - 0.32$ g $N_2O$-N kg$^{-1}$
$^{1}$ grain and $5.11 - 29.86$ g $N_2O$-N kg$^{-1}$ grain N, respectively; with significantly lower values for
CA and CA+BC than for Conventional in the first season. In the second season, treatments had
no effect ($p > 0.05$) on yield and yield scaled emissions. Maize yield and yield scaled emissions
in the conventional treatment (ConventMM) were significantly higher ($p < 0.05$) in the second
season compared to the first season.













4.0. Discussion
4.1. Dynamics of $N_2O$-N fluxes
Averaged daily $N_2O$ fluxes ranged from 1.0 – 51.2 µg N m$^2$ h$^{-1}$ (Fig 1a), and the cumulative
$N_2O$-N emissions from May 2023 – January 2024 were less than 1.2 kg $N_2O$-N ha$^{-1}$ (Fig 4, 5).
These ranges are consistent with other research carried out acidic tropical soils under
conservation agriculture with (Fungo et al., 2017; Munera-Echeverri et al., 2022) or without
biochar addition (Shumba et al., 2023). Low $N_2O$ emissions can be attributed to low soil
mineral N contents (Fig 1b, c) (Chapuis-Lardy et al., 2009). Higher $N_2O$-N fluxes were
recorded in October 2023 after harvesting pigeon pea and immediate sowing for the second
season (Fig 1a). These emissions might have been associated with decomposition of pigeon
pea residues, leaf litter and root turnover. This would indicate that crop residues were a source
of C and N substrates that induced significant $N_2O$ production. Additionally, consumption of
labile C by heterotrophs might have created anaerobic microsites that promoted denitrification,
especially as this period coincided with heavy rainfalls and high WFPS values (Fig. 2d). A
similar, but smaller emission peak was seen in June when abundant rainfalls terminated a dry
spell. Rewetting of dry soil triggers increased $N_2O$ emissions likely due to increased
nitrification and denitrification fueled by release of readily available N and C from dead
microbial biomass (Namoi et al., 2019). Likewise, after harvest and sowing, plant N uptake is
small, which might have supported elevated microbial C and N turnover. $N_2O$ emission peaks
were short-lived lasting for only 2 weeks (Fig 1a).

4.2. Mineral N and WFPS
Daily $N_2O$ emissions were weakly corelated to mineral N. The Gulu site has a relatively low
soil $\delta^{15}N$ value of 4.64 and a soil N content of 0.11% (Namatsheve et al., 2025), indicating a





highly efficient and tight N cycling (Craine et al., 2015).This could mean that microbes
compete well for mineral N in these soils, probably immobilizing available N rather than
releasing it for microbial N transformations like nitrification and denitrification. We anticipated
that biological $N_2$-fixation by pigeon pea in the first season would result in higher $N_2O$
emissions in the second season, especially in CA treatments where $N_2$-fixation was high
(Namatsheve et al., 2025). However, $N_2O$ fluxes and mineral N did not appear to be driven by
this $N_2$-fixation. Our results imply that the process of symbiotic N fixation *per se*, and residue
retention do not affect soil mineral N and $N_2O$ emissions in unfertilized soils with inherently
low N. Rochette et al. (2004) also reported that there is considerable uncertainty related to the
emissions of $N_2O$ from soils under legumes, and the soil mineral N alone was a poor indicator
of $N_2O$ emissions for two seasons in acidic soils in Canada.
Generally, $NH_4^+$ and $NO_3^-$ contents were more variable in the first season (May – October) than
the second season (October – January) (Fig 1b, 1c, 3). At the onset of the experiment, mineral
N was most likely from mineralisation of chemically mulched grasses having grown on the
fallow for 3 years prior to the experiment. Across all seasons, significantly higher $NH_4^+$ was
recorded in CA systems than conventional treatments, mainly due to mineralisation of pigeon
pea crop residues. Biochar amendments in CA systems did not affect $NH_4^+$ and $NO_3^-$. Our result
for mineral N aligns with the findings of Munera-Echeverri et al. (2022), who reported
similarly low concentrations of $NO_3^-$ (0.8 – 4 mg kg$^{-1}$) and $NH_4^+$ (4.3 – 10 mg kg$^{-1}$) in a fertilised
Arenosol in Zambia.
High $N_2O$ emission rates go often along with high WFPS values, increasing the anaerobic
volume and hence denitrification in soils (Hao et al., 2025; Wang et al., 2023). Our results
indicated a weak positive relationship between WFPS and $N_2O$ emissions ($R^2 = 0.02$, $p < 0.001$;
Fig. 7). High evaporation under tropical conditions due to high temperatures (mean 25 °C,
range 23 – 34 °C) results in rapid water loss, which drastically reduced the time the soil would



be above 60% WFPS, despite high rainfalls throughout the sampling period (Fig 1d, e). Apart
from June – July 2023, WFPS was <60%, which is often considered a threshold for
denitrification-driven $N_2O$ emissions. This may, in part, explain the weak correlation between
WFPS and $N_2O$ fluxes in our study.

4.3. Effects of crop rotation, tillage and biochar on $N_2O$ emissions
Cumulative $N_2O$ emission in the conventionally tilled pigeon pea–maize rotation
(Conventional) did not differ from the conventionally tilled maize monocrop (ConventMM)
(Fig 5, Fig S3). The trials were established in a soil with low organic N content without
applying N fertilizer. This is in line with Phiri et al. (2025) who did not find any effect of
conservation agriculture on soil N over a short period of time. However, after some seasons,
nutrient cycling, inputs from crop residues and biological $N_2$-fixation might enhance soil
fertility. Jeuffroy et al. (2013) reported a 75 – 80% reduction in $N_2O$ emissions in a 4-year study
under pea rotation without N input compared to a fertilized monocrop, illustrating the
significance of N input for $N_2O$ emissions, rather than the effect of rotation itself. We did not
find the effect of CA (reduced tillage) on $N_2O$ emissions. However, Jantalia et al. (2008) and
Ruan and Philip Robertson (2013) reported higher $N_2O$ emissions under conventional tillage
compared to reduced tillage, which they attributed to soil disturbance, increased soil aeration
and accelerated organic matter breakdown.
Biochar applied at a rate of 4 Mg ha$^{-1}$ in planting basins in the CA system did not affect $N_2O$
emissions. In theory, biochar with a high C:N ratio of >60, as applied in this study, could reduce
the bioavailability of inorganic N through microbial immobilisation (Namoi et al., 2019) or
sorption of $NO_3$ due to unconventional H-bonding between $NO_3^-$ ions and biochar surface
functional groups (Kammann et al., 2017; Nguyen et al., 2017). However, these mechanisms



appear to be relevant for $N_2O$ emissions only when inorganic N fertilizers are applied. This is
a newly established experiment, and the treatment effects may not have manifested yet. In a
similar short-term study, Munera-Echeverri et al. (2022) reported that biochar amendments did
not affect $N_2O$ emissions despite increased gross nitrification rates in the biochar treatments.
Contrary, biochar was shown to effectively reduce $N_2O$ emissions in long-term studies where
inorganic N fertilizers were applied. Fungo et al. (2019) reported a 22% reduction in emissions,
over three years in a fertilized Ultisol in western Kenya. Similarly, Case et al. (2015) reported
that biochar suppressed $N_2O$ emissions in a sandy loam soil fertilized with 140 kg N ha$^{-1}$ yr$^{-1}$,
over a period of 3 years. Lentz et al. (2014) also found that biochar reduced $N_2O$ emissions by
50%, indicating that biochar inhibited nitrification and N immobilisation.

4.4. Yield and yield scaled $N_2O$ emissions
Treatments had no significant effect on grain and grain N yields in either season. Again, as
these were newly established trials, we did not expect to see immediate effects of CA and
biochar amendment. Rusinamhodzi et al. (2011) highlighted that high inputs of especially N
fertilizer are required to realize yield benefits of CA. When inorganic fertilizer is applied,
positive effects of CA become more prominent in the long term. In Zambia, benefits of biochar
and/or CA on grain yield were reported after several seasons (Martinsen et al., 2014, 2017;
Munera-Echeverri et al., 2020). Yields in the first season under conventional tillage with maize
monocropping (ConventMM) were low, <1.6 t ha$^{-1}$ the average maize yield in Uganda without
N fertilization (Kaizzi et al., 2012). Low rainfall received from early May to mid-June, during
the critical growth stage for maize (tasselling and grain filling) drastically reduced maize grain
yield in the first season.



Reducing yield-scaled $N_2O$ emission has been pointed out as one of the most promising
strategies to increase crop yield while reducing $N_2O$ emissions. In this study, yield-scaled $N_2O$
emission were $0.16 – 0.32$ g N kg$^{-1}$ grain in the first season, and $< 0.20$ g N kg$^{-1}$ grain in the
second season. During the first season, yield-scaled $N_2O$ emission in CA and CA+BC were
significantly reduced by 50 % compared to conventional practices, indicating that N use
efficiency was high. These practices were effective in minimising emissions without penalising
pigeon pea productivity, supporting CA as a sustainable agricultural practice. In the second
season, yield scaled emissions were low, but no treatment effect was recorded. Our results are
in line with other studies in SSA, although they applied mineral N fertilizer; for example,
Shumba et al. (2023) reported yield scaled emissions of $0.09 – 0.19$ in maize after applying 58
kg N ha$^{-1}$ in a Ferralsol and Lixisol in Zimbabwe.

5.   Conclusions
$N_2O$ emissions were not affected by biochar addition in planting basins under CA systems,
likely because in low-input systems without fertilization microbial immobilization prevails
over the influence of biochar on mineral N availability. We established that $N_2O$ emissions
peaks following rainfall events after dry spells and the incorporation of high-quality pigeon pea
residues were short lived, indicating that residue management may have temporary effects on
$N_2O$ emissions in unfertilized systems. Yield scaled $N_2O$ emissions were substantially lower
under CA and CA+BC systems, implying that $N_2O$ emissions can be reduced without
penalising pigeon pea grain yield.





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
