# Peer review of "Nitrous oxide emission from pigeon pea maize rotation in response to conservation"

_EGUsphere, 2025_

## Referee Comment (RC2)

The authors have conducted an experiment in Gulu, Uganda in which the effects of various conventional and conservation agricultural treatments on crop productivity, soil N dynamics, and N2O emissions are analyzed. The study is conducted over two cropping seasons within a single year – April – October 2023 and October-Jan 2024. Plots that had been fallow for the previous 3 years were cleared and prepared in an RCB design with four treatments: conventional maize monocrop (ConMM), conventional pigeon pea/maize (Con) rotation, conservation agriculture (CA), and conservation agriculture with biochar ammendments (CA+bc). Some differences were observed in NH4 but not in NO3. Maize grain yield in season 1 was lower in conMM than pigeon pea grain yield in the other treatments; there was no difference in grain yields in the second season. Yield-scaled N2O emissions were higher from the conventional than the CA treatments in the first season, but there were no differences in the second season.

This study adds much-needed measurements of N cycling and N2O emissions from low/no-input cropping systems in Uganda, and I look forward to seeing them in the peer-reviewed literature. I do have several general comments.

Major comments:

A major limitation of this study is that data are only presented from a single year (which encompasses a long-rain and a short-rain cropping season). Interannual variability can play a big role in crop productivity and N cycling (highlighted by the statement in lines 548-550 that drought drastically reduced maize grain yield in the first season), and this type of study would benefit from more than one year of measurements. It is true that two cropping seasons are included, but these are different cropping seasons – long rains and short rains—and cannot be said to represent the same thing as multiple years of measurements. The fact that there are only one year of data presented here and, as a result, it is not really possible to generalize really needs to be emphasized front-and-center in the text. (There are so many examples in ecological studies of a second year of data drastically changing results that it's become cliché to joke that "I should have quite collecting data after one year!") That said, 1 year of data can still be quite useful, and I think it's particularly important to have these data published because there are so few data from the region, and particularly few data from pigeon pea systems. And I am especially sympathetic to the tremendous amount of effort required to establish these experimental plots and to collect and analyze these data. I leave it to the editor to decide whether a revised manuscript would meet biogeosciences' editorial threshold for publication.

On a somewhat related note, the authors' suggestion that treatment differences were not expected in this first year (e.g., Line 540-542) would seem to raise a fundamental question: if treatment differences were not expected, then why were measurements conducted in the first year, and why is it interesting to publish them? A way to address this concern would be to cite examples from the literature where treatment differences (when similar treatments were compared) *were* found in the first year of establishment.

**Other general comments:**

I don't understand why chamber position is included as a treatment in statistical analyses. If there were some targeted analyses into mechanism then it might make sense, but in this manuscript the central questions are how the agricultural management treatments differ, and how soils change over time within an agricultural management treatment. To answer this question, the weighted mean of the within row and interrow chambers would be the appropriate response variable—i.e., an estimate of the flux from each plot.

I think there's a missed opportunity for more discussion of what makes this study unique-- the focus on unfertilized systems. Currently this is not discussed much at all. There are actually quite a lot of N2O data from unfertilized maize in tropical African systems that could provide context for understanding the emissions in this study– the control plots from fertilizer trials. Off the top of my head, there are studies in East Africa with lead authors that include Baggs, Hickman, Millar, Pelster, Rosenstock, Tully, Zheng, and studies in Zimbabwe from Mapanda. Typically there are multiple studies from each author, and at least some of them include explicit calculations of yield-scaled emissions.

Initial soil sampling for NO3- and NH4 analyses was conducted after treatment application. Since there were no NO3- and NH4+ measurements before treatment, these measurements are able to answer one question: how treatment affects changes in NO3- and NH4+ snap shots when comparing the start of the first crop season to the end of the second crop season of a given year. I think it would be very useful to include some discussion/contextualization as to why this is an important question—it's not immediately clear to me.

**Specific comments:**

Line 40: delete "tropospheric"

Line 42: N2O is actually the largest anthropogenic driver of stratospheric ozone depletion (see Ravishankara 2009 in Science DOI: 10.1126/science.1176985)

Line 44: replace 'atmospheric' with 'anthropogenic' and change to "60% to global **anthropogenic** N2O emissions"

Line 60: you can delete 'or by-product'

Line 64: also labile carbon availability ('substrate availability' would most likely be interpreted to refer to nitrogen compounds)

Line 137: indicate in the text when specifically the clearing was conducted and when glyphosate was applied.

Line 138: I'm not an expert in herbicides, but I think glyphosate can potentially have a range of impacts on crop production ( via uptake by crops either from glyphosate that has persisted in soil or is released from decomposing weed biomass; it also has potential interactions with mineral nutrition: see https://pmc.ncbi.nlm.nih.gov/articles/PMC6918143/), which makes the timing of its use useful to know, and which might have implications for season 1 production.

Line 139: indicate in the text specifically when the plots were tilled

Line 144: ConventMM has not been defined at this point. The CA+BC abbreviation has also not been introduced—it was introduced as CA+biochar.

Line 157-158: I'm a little confused by plant spacing. In line 141, it sounded like the spacing was 70cm x 35 cm, which would yield a planting density of 40,898. Here it says 10cm; if that's 10cm x 70cm, that would be a planting density of 143,143. And were the same planting densities really used for maize and pigeon pea?

Line 175-176: two different biochar treatments are described here, including one that isn't defined or mentioned anywhere else in the manuscript (CA+BC+BC).

Line 213: Be specific about when the collars were installed (you can say something like 'at least xx days before the first measurement).

Line 210-211: I don't think I quite have a complete description of the chamber top—it sounds like it should be a pvc pipe – is it a pipe that was manufactured with a top, or did you have to add and seal a top to the pipe?

Line 218: citing a reference here for using petroleum jelly would be great. It sounds like it would work, but a reference would be helpful.

Line 220: were the initial measurements really taken at 1 minute? (i.e. sampling was delayed until the chamber had been sealed for a full 60 seconds)

Line 220: 60 minutes is a fairly long period. Can you establish that N2O did not reach or approach equilibrium in the chamber?

Line 228: Did you have any checks to make sure there wasn't any sample contamination/leakage during shipping? (e.g., having vials filled with a standard gas in Uganda and then measured in Norway.)

Line 235: how was it determined when a linear or when a polynomial fit should be used? (i.e., it would be good to make sure overfitting was avoided; the additional parameter in the polynomial model can provide a closer fit to the data even if it is inferior to the linear model)

Line 242: a quadratic fit does not seem like an appropriate way to calculate a flux here, and makes me a bit worried about the polynomial fits in general. If concentrations are convex downward, that suggests to me that concentrations in the chamber have equilibrated with soil over the 60-minute period, and the resulting flux calculated would not be a good estimate of soil fluxes in the absence of a sealed chamber. If polynomials are fit to data in for which the slope between 30 and 60 minutes is smaller than the slope between 1 and 15 minutes, again it sounds like you're seeing a chamber artifact where concentrations in the chamber are approaching equilibrium with concentrations in the soil, and you might be better off using data from 1 to 30 minutes.

Line 249: Instead of 'scaling up' I think it would be more precise to say that you're estimating a flux that is representative of the entire plot, and it's being done by calculating a weighted mean of fluxes from the basin and interrow chambers. You could write something like "We estimated a flux representative flux for each plot ($N_2O_{plot}$) by calculating a weighted mean of fluxes from the basin and interrow chambers. We used weighting factors of 0.12 for basin . . .".

Line 259: indicate when extractions were conducted relative to sampling (e.g., x hours, the next day, etc)

Line 280: I would change "dividing the scaled cumulative N2O emissions . . ." to something like "dividing the weighted mean N2O emissions" or "dividing $N_2O_{plot}$ emissions. . ."

Line 281: I would change "scaling factor" to "weight"

Line 283: I would change "scaling factor" to "weight"

line 293, you indicate that random effects were introduced to account for repeated measurements. I need more justification for this: typically, one assumes that repeated measures of the same plot will be correlated to some extent. And since statistics are only conducted on cumulative variables (which is perfectly reasonable, and a good way to avoid issues of autocorrelation in repeated measures), I'm not sure why time is discussed as an effect at all.

Line 296-298: I find it rather remarkable that no data transformations were required (I've never seen field data like this that didn't).

Line 333: change "were" to "when"

Line 349: change "second respectively" to "second season, respectively"

Figure 2 & 3: I think you want to combine these into a single figure—I don't think it's necessary to compare Conv to convMM in one figure and then compare Conv to the CA treatments in a separate figure, especially since you're presenting post-hoc contrasts.

The higher nH4+ in convMM than all other treatments in season 1 seems unexpected, and (maybe) higher or equivalent to the CA pigeon pea treatments in both years? Something that may be worth discussing.

Line 388: Delete the text "Fig. 4" -- it does not present cumulative emissions

Line 392: Interpreting the treatment main effect in the presence of a significant treatment x position interaction (which says that the treatment effect depends on the chamber position) is complicated—another reason to use the weighted mean flux for each plot as the response variable. But I do believe interpretation of a significant main effect in the presence of an interaction in an RCB design does indicate a main effect that is over and above the treatment x block interaction.

Figures 5 and 6: I think you want to combine these into a single figure—I don't think it's necessary to compare Conv to convMM in one figure and then compare Conv to the CA treatments in a separate figure, especially since you're presenting post-hoc contrasts.

Table 3:

- It would be helpful to have a reminder in the Table caption that the weighted mean fluxes were used to calculate yield-scaled emissions.
- I would add something to the discussion explaining why we would care about N-yield scaled emissions.
- Why are the season comparisons (capital letters) only included for ConventMM and not the other treatments?
- I would explain in the table caption why ConventMM is not included in the treatment tests (lower case letters) in Season 1 (presumably because comparing maize yields to pigeon pea yield isn't a useful comparison).

- I think it would be useful to include the P values rather than "ns." It provides the reader with more information and context for interpreting the results. A P value of 0.08 is very different from a P value of 0.5

Line 434: Again, and throughout the manuscript, I would report the actual P value rather than P>0.05 or P<0.05

Line 434-443: I would just make it explicit here that you are not including ConvMM in treatment comparisons for season 1 since maize was grown in that treatment rather than pigeon pea, as in all the other treatments.

Line 442: "significantly higher" needs to be changed: Maize yield was higher, but yield-scaled emissions were lower

Line 461: I think it would be useful to compare emissions to conventional agricultural settings as well—control plots from other experiments in East Africa. It may also be useful to provide emissions—including yield-scaled emissions--from conventional management using fertilizer as context for the conventional and CA plots in this study, while also discussing issues associated with no-input agriculture such as nutrient depletion—how long could these different practices remain sustainable?

Line 472: https://doi.org/10.1029/2020JG005742 could be helpful for the rewetting discussion.

Line 487: much of the fixed N may be in the harvested biomass, limiting the amount returned to soil.

Line 494: I don't think NH4 is higher in CA systems than conventional systems—isn't the NH4 in ConvMM no different than the CA treatments? And if ConvMM is higher than conv, it doesn't follow that the mineralization of pigeon pea residues is responsible for the difference. It also conflicts with the statement in line 486.

Line 514-516: this sentence is missing something, or 'nutrient cycling' should be deleted.

Line 540-542: You may want to rephrase this.  Ff no treatment effects were expected, it would seem to undermine a justification for the entire experiment (i.e., readers may ask why measurements were conducted in a year when no treatment effect was expected, instead of at a time when the treatments would be expected to have an effect).

Line 566: delete "We established that"

---

## Author Comment (AC1)

Reviewer 1 _N2O manuscript

Response to review of the manuscript "Nitrous oxide emission from pigeon pea – maize rotation in response to conservation agriculture and biochar amendments in a Ferralsol, northern Uganda" by Namatsheve et al.

This study investigates the effects of conservation agriculture (CA), both with and without biochar amendments, compared to conventional tillage in an unfertilised maize-pigeon pea rotation in Uganda. It also compares these treatments with a maize monoculture treatment.

The manuscript significantly contributes to the scientific knowledge base by presenting a novel dataset.

The introduction shows the relevance of the study and the basis of existing literature. The study design and hypotheses are well described. The data is highly valuable. It contains 17 observations of $N_2O$ emissions and its driver variables per treatment over the course of one year (two seasons) in Uganda. The methods are sound and well described. The results are well described and the discussion a good start, but both need some revisions as suggested below.

I recommend to publish the manuscript after major revisions.

Response: We thank the reviewer for the constructive evaluation of our manuscript. We appreciate the recognition of the novelty and relevance of our dataset, the clarity of our study design and hypotheses, and the value of our methodological approach.

We have carefully addressed all specific comments and suggestions to improve the clarity and scientific rigor of the results and discussion sections, and we believe these revisions have strengthened the manuscript and hope it now meets the standards for publication in Biogeosciences.

L30: Overall, across all seasons, cumulative growing-season (279 days) $N_2O$ emissions ranged from 0.44 – 1.11 kg N $ha^{-1}$

RC: Since this is a major finding, you should give the values specifically per treatment in the abstract.

Response: We thank the reviewer for the comment, we have now added the values per treatment: "Overall, across two growing seasons, weighted cumulative $N_2O$ emissions in 279 days ranged from 0.46 kg N $ha^{-1}$ in CA+BC treatment to 0.88 kg N $ha^{-1}$ in Conventional treatment, respectively."

L127: Table S19

RC: You start with Table S19.   You should instead name your Tables differently and make sure to have them ordered in the same way as they occur in your text. You start with table S1 here and so on.

Response: Thank you for the comment, we have now rearranged the tables, and numbered them in logical order (viz. table S1).

L130: RC: Please add average precipitation sums (for example 5 or 10 years) in order to show how usual the 2023 conditions were.

Response: Gulu received annual rainfall of 1460 mm $yr^{-1}$ from 1980 – 2010 (Oriangi et al., 2024). The average annual temperature is 24 ℃, and the annual rainfall in 2023 was 1238 mm, of which 818 mm was received in the first and 419 mm in the second season. (Ln 132 – 135).

L135: Is it known how long the maize and cassava production have been practiced?

Response: We have included this information as follows: The experiment was established on a field that was fallowed for three years; before that it was used for maize and cassava production without fertilization, for at least five years (Ln 136 – 137).

L160: For CA, weeds were controlled by spraying glyphosate at a rate of 1.03 L ha-1, immediately after sowing and hand pulling throughout the season.

RC: Set in context how realistic glyphosate spraying would be under farmers practice conditions.

Response: Subsistence farmers normally control weeds with hand hoeing. Herbicide application is being promoted in the mainstream of conservation agriculture to reduce tillage and to control troublesome perennial weeds that are otherwise difficult to control with hand hoeing. Although the use of glyphosate among farmers is a relatively new technology, it is readily available at agro-dealers within the smallholder farming communities, and some farmers use it.

L 198: RC Write: Plot wise sampling in 0 – 20 cm was carried out at the onset (April 2023) and end (October 2023) of the first growing season from planting basins in CA

and CA+Biochar treatments and in the planting rows in conventional treatments to assess the effect of different treatments on soil properties.

Leave out the next sentence.

Response: Thank you for rephrasing L198. We have now deleted the next sentence

L234: by (Žurovec et al., 2017).

RC: write by Žurovec et al. (2017).

Response: Thank you for the comment, the reference is now correctly cited.

L246: $\sum(fi + (fi+1)/2 \times (ti+1 - ti) \times 24 \times 10-5$

RC: There is a bug in this formula, there are opening up three brackets but only two of them close again. Make sure to provide a correct and well formatted equation.

Response: Thank you for the comment, the formulae is now written as follows:

$$cumulative\ N_2O = \sum[(f_i + f_{i+1})/2 \times (t_{i+1} - t_i) \times 24 \times 10^{-5}] \qquad (2)$$

L255: (NO3-N and $NH_4^+$-N)

RC: Use $NO_3^-$ here and elsewhere (Text & Fig. to ensure consistency.

Response: Thank you, we have now changed this in the whole manuscript.

L281-283: Adjust the text, since it is the same as before:

RC: My suggestion: For calculating yield scaled N2O emissions, the same scaling factor as for cumulative N2O fluxes of 0.12 for basin and 0.88 for interrows in CA treatments, and 0.50 for inrows and interrows in conventional treatments were applied.

Response: We have now rephrased the sentence as follows: Yield-scaled $N_2O$ emissions (kg $N_2O$-N $kg^{-1}$ grain yield) and N-yield scaled emissions (kg $N_2O$-N $kg^{-1}$ grain N) were estimated for each season by dividing the area-weighted cumulative $N_2O$ emissions with grain yield or N content of the grain (N concentration × grain yield).

L297: RC: The statistical approach makes sense. Nevertheless, I wonder how residuals data can be normal without any transformation, this is very unusual for N2O, could you show the residual plot?

Response: For cumulative $N_2O$ emissions, we checked the normality of residuals. Visual inspection of the QQ plots showed that residuals were near normal without transformation as shown in fig 1.

[Figure]

Fig 1: Cumulative $N_2O$

For N2O fluxes, The QQ plot shows a deviation from normality in the tails, which is common for $N_2O$ flux data. However, mixed-effects models are robust to violations of normality in residuals (Zuur et al., 2009), especially with large sample sizes. We tested log transformation, but it did not improve model fit (AIC/BIC remained similar), so we retained the original scale for interpretability.

[Figure]

Fig 2: N$_2$O fluxes

[Figure]

Fig 3: Original N$_2$O fluxes vs log transformed N$_2$O fluxes

3.0 Result -> 3 Results

L316 : SOC ranged from 1.25 – 2.23% and biochar significantly increased C

RC: use SOC consistently since this is the abbreviation you introduced.

Response: Thank you for noting this; We now use SOC throughout the manuscript.

Figure 1: The insert in (a) shows mean ± se N2O fluxes during peak emission on 18 October 2023.

RC: This is the same as you see when looking at this peak day in the time series plot, so I do not see the additional value to much. Well if you want to pronounce it a lot you can keep it. I would rather show the cumulative values - but this might show up later in a separate Figure?

Response: we removed the insert in (a)

Table 3: Please round the yields meaningfully (Consider the accuracy of measurements!). Give no decimal number here if you have kg /ha. Same for N yields.

Response: Thank you for the comment, we have now rounded off the grain yield and N yield to whole numbers.

Figure 5+ 6 : Add the **weighted cumulative** values to the plots as a third colour besides basin/inrow and interrow, since this is the main result that should be also show here.

Indicate Tukey-HSD letters also in Fig 5

Response: Thank you for the comment. We have now included the weighted cumulative values for Fig 4 and 5. We also added letters to show significant differences in Fig 5 (Ln 397 – 413).

L388: Write two sentences, else it is confusing: Chamber position did not significantly explain variations in cumulative N2O emissions. There were no significant differences (p > 0.05) between conventionally tilled pigeon pea – maize rotation and conventionally tilled continuous maize monocropping (Table S17, S18, Fig. 5, Fig S3)

 Response: Corrected

L477: corelated -> correlated: Please check spelling (and grammar) throughout the manuscript- I didn't.

Response: corrected

L481: probably immobilizing available N rather than releasing it for microbial N transformations; RC: I know what you mean but... The phrase "releasing it for microbial N transformations" is a bit unclear — microbes don't exactly "release" N for other microbes; it may be better to phrase it differently:

Response: Thank you for a helpful suggestion. The sentence is now rephrased as follows: This suggest that microbes in these soils compete effectively for mineral N, likely immobilizing it and thereby reducing its availability for microbial N transformations such as nitrification and denitrification.

L485: Our results imply that the process of symbiotic N fixation per se, and residue

retention do not affect soil mineral N and N2O emissions in unfertilized soils with inherently low N. Rochette et al. (2004) also reported that there is considerable

uncertainty related to the emissions of N2O from soils under legumes, and the soil mineral N alone was a poor indicator of N2O emissions for two seasons in acidic soils in Canada.

RC: 1) It is well-known that N2 fixation itself does not affect N2O emissions, please rephrase so this gets clear.

2) In many studies soil mineral N is a bad indicator for N2O  emissions. This is nothing particular to legumes. Please adjust.

Response: Thank you for the insightful comments. We agree that there is no direct link between $N_2$-fixation and $N_2O$ emissions. We also acknowledge that soil mineral N is a poor indicator of $N_2O$ emissions. We have now rephrased as follows:

Our results imply that $N_2$-fixation and residue retention do not directly affect soil mineral N or  $N_2O$ emissions in unfertilized soils with inherently low N. Rochette et al. (2004) also reported considerable uncertainty in $N_2O$ emission from soils under legumes, and they noted that soil mineral N alone was a poor predictor of $N_2O$ emissions for two seasons in acidic soils in Canada.

L536: Lentz et al. (2014) also found that biochar reduced N2O emissions by 50%, indicating that biochar inhibited nitrification and N immobilisation.

RC: Did they measure nitrification and immobilisation? It is a bit contrary to the study by Munera-Echeverri et al. (2022), so the effect of biochar on nitrification and the difference between these studies should be explained.

Response: Lentz et al., 2014 measured N2O emissions and net mineralization, which is affected by nitrification and N immobilisation. We agree that Lentz et al., 2014 contradicts Munera-Echeverri et al., 2022 results. We have rephrased the whole paragraph, and we have now discussed the weighted N2O emissions results.

Biochar applied at a rate of 5 Mg ha$^{-1}$ to CA systems reduced weighted $N_2O$ emissions by 33% and 66% compared to Conventional treatment in the first and second season, respectively. Biochar with a high C:N ratio of >60, as applied in this study, was shown to reduce the bioavailability of inorganic N through microbial immobilisation (Namoi et al., 2019) or sorption of $NO_3^-$ due to unconventional H-bonding between $NO_3^-$ ions and biochar surface functional groups (Kammann et al., 2017; Nguyen et al., 2017). In a fertilized Ultisol in western Kenya, Fungo et al. (2019) reported a 22% reduction in emissions. Case et al. (2015) also reported that biochar suppressed $N_2O$ emissions in a sandy loam soil fertilized with 140 kg N ha$^{-1}$ yr$^{-1}$. Biochar applied in a calcareous soil also reduced $N_2O$ emissions by 50%, and increased soil $NH_4^+/NO_3^-$ ratio, indicating that biochar impaired nitrification and N immobilisation processes (Lentz et al., 2014). Our results support the growing

evidence that biochar can mitigate $N_2O$ emissions via various N cycling modifications (Liu et al 2018, Zhang 2021, Borchard et al., 2018; Kammann et al., 2017). However, in a short-term study, Munera-Echeverri et al. (2022) reported that biochar amendments did not affect $N_2O$ emissions in Zambia despite increased gross nitrification rates in the biochar treatments. In a global meta-analysis, Shakoor et al. (2021) showed that biochar increased $N_2O$ emissions by 20%. Discrepancies on response of $N_2O$ emissions to biochar might be explained by biochar type, soil parameters, climatic conditions and experimental duration. Yield and yield scaled $N_2O$ emissions. (Ln 537 – 555).

L554 – 561:

End the paragraph with your study, not with the numbers of another study. Put the following sentence to line 554: Our results are in line with other studies in SSA, although they applied mineral N fertilizer; for example, Shumba et al. (2023) reported yield scaled emissions of 0.09 – 0.19 in maize after applying 58 kg N ha-1 in a Ferralsol and Lixisol in Zimbabwe.

Rephrase to make it consistent again

Then end the discussion with this: These practices were effective in minimising emissions without penalising pigeon pea productivity, supporting CA as a sustainable agricultural practice.

Response: Thank you for the useful comment we have taken the comment into consideration (Ln 573 – 575).

L569: Typo: Yield-scaled

Response: corrected

---

## Author Comment (AC2)

Reviewer 2

The authors have conducted an experiment in Gulu, Uganda in which the e9ects of various conventional and conservation agricultural treatments on crop productivity, soil N dynamics, and N2O emissions are analysed. The study is conducted over two cropping seasons within a single year – April – October 2023 and October-Jan 2024. Plots that had been fallow for the previous 3 years were cleared and prepared in an RCB design with four treatments: conventional maize monocrop (ConMM), conventional pigeon pea/maize (Con) rotation, conservation agriculture (CA), and conservation agriculture with biochar ammendments (CA+bc). Some di9erences were observed in NH4 but not in NO3. Maize grain yield in season 1 was lower in conMM than pigeon pea grain yield in the other treatments; there was no di9erence in grain yields in the second season. Yield-scaled N2O emissions were higher from the conventional than the CA treatments in the first season, but there were no di9erences in the second season.

This study adds much-needed measurements of N cycling and N2O emissions from low/no-input cropping systems in Uganda, and I look forward to seeing them in the peer reviewed literature. I do have several general comments.

Response: We thank the reviewer for the constructive evaluation of our manuscript. We appreciate the recognition of relevance of our study.

We have carefully addressed all specific comments and suggestions to improve the clarity and scientific rigor of the results and discussion sections, and we believe these revisions have strengthened the manuscript and hope it now meets the standards for publication in Biogeosciences.

Major comments:

A major limitation of this study is that data are only presented from a single year (which encompasses a long-rain and a short-rain cropping season). Interannual variability can play a big role in crop productivity and N cycling (highlighted by the statement in lines 548550 that drought drastically reduced maize grain yield in the first season), and this type of study would benefit from more than one year of measurements. It is true that two cropping seasons are included, but these are di9erent cropping seasons – long rains and short rains—and cannot be said to represent the same thing as multiple years of measurements. The fact that there are only one year of data presented here and, as a result, it is not really possible to generalize really needs to be emphasized front-and-center in the text. (There are so many examples in ecological studies of a second year of data drastically changing

results that it's become cliché to joke that "I should have quite collecting data after one year!") That said, 1 year of data can still be quite useful, and I think it's particularly important to have these data published because there are so few data from the region, and particularly few data from pigeon pea systems. And I am especially sympathetic to the tremendous amount of e9ort required to establish these experimental plots and to collect and analyze these data. I leave it to the editor to decide whether a revised manuscript would meet biogeosciences' editorial threshold for publication.

Response: We agree with the reviewer that data from several subsequent seasons would have been valuable. However, due to budget and time constraints, this was not feasible. Nevertheless, we believe that presenting effects of different management practices on $N_2O$ emissions from an unfertilized system in two growing seasons is of interest to both the scientific community and stakeholders. Since the effects of biochar on soil chemical properties are expected to diminish over time (e.g., Cornelissen et al. 2018), measuring emissions immediately after biochar application for one full year provides meaningful insights. Therefore, we argue that our approach is justified.

On a somewhat related note, the authors' suggestion that treatment di9erences were not expected in this first year (e.g., Line 540-542) would seem to raise a fundamental question: if treatment di9erences were not expected, then why were measurements conducted in the first year, and why is it interesting to publish them? A way to address this concern would be to cite examples from the literature where treatment di9erences (when similar treatments were compared) *were* found in the first year of establishment.

Response: We agree that the sentence did not fit in, we have deleted it. We have now included area-weighted $N_2O$ emissions in the result section and the treatment effect is now clear, and we have now discussed the treatment effects.

**Other general comments:**

I don't understand why chamber position is included as a treatment in statistical analyses. If there were some targeted analyses into mechanism then it might make sense, but in this manuscript the central questions are how the agricultural management treatments di9er, and how soils change over time within an agricultural management treatment. To answer this question, the weighted mean of the within row and interrow chambers would be the appropriate response variable— i.e., an estimate of the flux from each plot.

Response: Chamber position was included as a factor in the statistical analysis to assess potential differences between basins/inrow where maize or pigeon pea was planted, also where biochar was applied) vs. interrow (no planting, no soil disturbance under CA, but overall digging under conventional). According to Abiven et al, (2015), root biomass inside planting basins was twice as large for biochar amended plots compared to regular CA in Zambia. In this unfertilized system, the main difference between basins and interrow location is that basins are re-opened every season under CA and that biochar is added to the basins only. Effects of biochar addition on soil properties were expected in the basins where biochar was applied, not the interrows, this influences emissions.

In addition to reporting differences between basins/inrow and interrow positions, we have now included the weighted cumulative N2O emissions in Fig 4 and 5, as well as yield- and yield-N scaled cumulative N2O emissions (Table 3) to show estimated total fluxes for the different management practices, -weighted for basins/inrow and interrow positions.

I think there's a missed opportunity for more discussion of what makes this study unique-the focus on unfertilized systems. Currently this is not discussed much at all.  There are actually quite a lot of N2O data from unfertilized maize in tropical African systems  that could provide context for understanding the emissions in this study– the control plots from fertilizer trials.  O9 the top of my head, there are studies in East Africa with lead authors that include Baggs, Hickman, Millar, Pelster, Rosenstock, Tully, Zheng, and studies in Zimbabwe from Mapanda.  Typically there are multiple studies from each author, and at least some of them include explicit calculations of yield-scaled emissions.

 Response: We agree with the reviewer that highlighting the uniqueness of our study in the discussion of our study, as it focuses on unfertilized systems under CA with legumes and biochar is essential. However, field studies in SSA reporting $N_2O$ emissions in CA systems with applications of at least 20 kg N ha$^{-1}$. This is based on our recent meta-analysis paper where we reviewed 26 studies specifically on N2O emissions in SSA (Namatsheve et al., 2024), see the figure below.

[Figure]

We have included this information in the manuscript as follows: In our recent meta-analysis of CA and biochar effects on N cycling, we found that residue retention increased soil $NO_3^-$, leading to higher $N_2O$ emissions; this finding was based on 26 published studies focusing specifically on $N_2O$ emissions, and applying at least 23 kg N ha$^{-1}$ (Ln 104 – 106).

Initial soil sampling for NO3- and NH4 analyses was conducted after treatment application. Since there were no NO3- and NH4+ measurements before treatment, these measurements are able to answer one question: how treatment a9ects changes in NO3- and NH4+ snap shots when comparing the start of the first crop season to the end of the second crop season of a given year. I think it would be very useful to include some discussion/contextualization as to why this is an important question—it's not immediately clear to me.

Response: We do not have the background values of mineral N. The absolute amounts especially for the first season are mainly influenced by the mineralisation of organic matter, built up during the 3 year fallow before the establishment of the experiment. The amount of NH4 and NO3 in the soil was investigated as potentially important explanatory variables for $N_2O$ emission. However, we still believe that comparison between treatments are valid. We have now included discussion on treatment differences.

**Specific comments:**

Line 40: delete "tropospheric"

Response: deleted

Line 42: N2O is actually the largest anthropogenic driver of stratospheric ozone depletion
(see Ravishankara 2009 in Science DOI: 10.1126/science.1176985)

Response: Thank you for the comment, we have now included the provided reference.

Line 44: replace 'atmospheric' with 'anthropogenic' and change to "60% to global **anthropogenic** N2O emissions"

Response: done

Line 60: you can delete 'or by-product'

Response: In nitrification, $N_2O$ is a by-product.

Line 64: also labile carbon availability ('substrate availability' would most likely be interpreted to refer to nitrogen compounds)

Response: Thank you for the comment. The sentence now reads as follows: The biogeochemistry of $N_2O$ in soil is to a large extent regulated by complex interactions between environmental and biogeochemical factors such as temperature, water, labile carbon availability, oxygen levels, acidity and substrate availability (Case et al., 2015; Tian et al., 2020).

Line 137: indicate in the text when specifically the clearing was conducted and when glyphosate was applied.

Response: the sentence now reads as follows: Prior to the establishment of the experiment, on 15 March 2023, a dense, naturally grown vegetation of grasses and shrubs was removed by slashing and chemical weeding using glyphosate [N-(phosphonomethyl) glycine]. (Ln 138 – 141).

 I'm not an expert in herbicides, but I think glyphosate can potentially have a range of impacts on crop production ( via uptake by crops either from glyphosate that has persisted in soil or is released from decomposing weed biomass; it also has potential interactions with mineral nutrition: see https://pmc.ncbi.nlm.nih.gov/articles/PMC6918143/), which makes the timing of its use useful to know, and which might have implications for season 1 production.

Response: Thank you for the comment, we appreciate your reference to relevant literature. In this study, glyphosphate was applied on 15 March 2023, 19 days prior to sowing for season 1 (4 May). This short period between spraying and sowing might not have be enough to degrade the chemical. Although, glyphosate remaining in soil can potentially have a range of impacts on crop production and may have potential interactions with mineral nutrition, its residual concentration has not been monitored after its application. Despite the potential challenges, glyphosate has been promoted in the mainstream of conservation agriculture where controlling perennial weeds with mechanical and biological methods is a challenge especially in early years of adoption.

Line 139: indicate in the text specifically when the plots were tilled

Response: Date for tillage is now included as follows:

On 27 March 2023, plots under conventional management were prepared by overall digging using hand hoeing (100% tillage) and plots of the same size under CA by manually digging 10-L planting basins (35cm long × 15cm wide × 20cm deep) spaced 70 cm × 35 cm (interrow × within row spacing). (Ln 141 – 144).

Line 144: ConventMM has not been defined at this point. The CA+BC abbreviation has also not been introduced—it was introduced as CA+biochar.

Response: Thank you for the comment. We have now changed CA+biochar to CA+BC in the whole manuscript, and we have now introduced ConventMM in the abstract and introduction.

Line 157-158: I'm a little confused by plant spacing. In line 141, it sounded like the spacing was 70cm x 35 cm, which would yield a planting density of 40,898. Here it says 10cm; if that's 10cm x 70cm, that would be a planting density of 143,143. And were the same planting densities really used for maize and pigeon pea?

Response: Plant densities for both maize and pigeon pea is the same. We accounted for the size of the basin which is 35cm long x 15cm wide x 20cm deep. Therefore, if the spacing for the basins is 70cm x 35 cm, and in each basin with 3 seeds spaced 10cm, one ha has 21 000 basins and a planting population of 63 000 plants ha$^{-1}$.

Line 175-176: two di9erent biochar treatments are described here, including one that isn't defined or mentioned anywhere else in the manuscript (CA+BC+BC).

Response: Thank you for noting this. We have now deleted CA+BC+BC, as it is not part of this paper .

Line 213: Be specific about when the collars were installed (you can say something like 'at least xx days before the first measurement).

Response: Thank you for the comment, we have now rephrased the sentence as follows: Permanent gas sampling plots were established by inserting 17 cm diameter PVC rings (the base) to a depth of 7 cm into the soil on 19 April 2023, 3 weeks before first sampling on 10 May 2023.

Line 210-211: I don't think I quite have a complete description of the chamber top—it sounds like it should be a pvc pipe – is it a pipe that was manufactured with a top, or did you have to add and seal a top to the pipe?

Response: The chamber top is a sewer pipe lid, check the figure below, also Fig S2. On top of the lids, we drilled holes and fit in the self-sealing rubber septum for gas sampling.

[Figure]

[Figure]

Line 218: citing a reference here for using petroleum jelly would be great.  It sounds like it would work, but a reference would be helpful.

Response: We have now included the citation as follows: To facilitate chamber deployment, the contact area between the collar and chamber was sealed with a thin layer of petroleum jelly as described by Shumba et al., (2023).

Line 220: were the initial measurements really taken at 1 minute? (i.e. sampling was delayed until the chamber had been sealed for a full 60 seconds)

Response: Yes during the first minute, in addition to sealing the chamber, the plunger of the syringe was pumped three times in and out before a representative sample was obtained.

Line 220: 60 minutes is a fairly long period. Can you establish that N2O did not reach or approach equilibrium in the chamber?

Response: We measured concentration in the chambers after 1, 15, 30 and 60 minutes. The 60-minute period was chosen because we anticipated relatively low $N_2O$ emission rates. In some cases, chamber N2O kinetics indicated that equilibrium was approached, i.e. convex downwards kinetics. In these cases we applied quadratic fits and reported the "time-zero" rate as given by the coefficient of the non-quadratic term, which comes close to estimating the flux from the first two measurement points.

We established that in most cases $N_2O$ did not reach equilibrium in the chamber. This is stated in the manuscript (Ln 232 – 234) as follows: Plotting measured $CO_2$ and $N_2O$ concentrations over time, revealed linear increase in most cases with little saturation observed. In some cases, the $N_2O$ concentration in the sample taken right after chamber deployment was substantially higher than 0.340 ppm (ambient $N_2O$ concentration), pointing at residual $N_2O$ in the chamber. The exalted $N_2O$ concentration after chamber deployment usually decreased until the second measurements (15 min) and to avoid fitting negative fluxes, the first sampling point was discarded. Flux rates were estimated by fitting a linear or second order (polynomial) function to the concentration change over time. A quadratic fit was only used in few cases in which $N_2O$ accumulation in the chamber showed a convex downwards trend, i.e., decreasing emissions. (Ln 229 – 237).

Line 228: Did you have any checks to make sure there wasn't any sample contamination/leakage during shipping? (e.g., having vials filled with a standard gas in Uganda and then measured in Norway.)

Response: He-filled vials were included as blanks to check for contamination during storage and shipment of the vials. Detected concentrations of $CO_2$ and $N_2O$ were

<5% of ambient. We used slightly over-pressured, crimp sealed glass vials with thick butyl septa, which have been shown previously to maintain pressure and mixing ratios during air transport and storage (Raji and Dörsch, 2020). (Ln 220 - 224).

Line 235: how was it determined when a linear or when a polynomial fit should be used? (i.e., it would be good to make sure overfitting was avoided; the additional parameter in the polynomial model can provide a closer fit to the data even if it is inferior to the linear model)

Response: Flux rates were estimated by fitting a linear or second order (polynomial) function to the concentration change over time. A quadratic fit was only used in few cases in which $N_2O$ accumulation in the chamber showed a convex downwards trend, i.e., decreasing emissions. (Ln 234 – 237).

Line 242: a quadratic fit does not seem like an appropriate way to calculate a flux here, and makes me a bit worried about the polynomial fits in general. If concentrations are convex downward, that suggests to me that concentrations in the chamber have equilibrated with soil over the 60-minute period, and the resulting flux calculated would not be a good estimate of soil fluxes in the absence of a sealed chamber. If polynomials are fit to data in for which the slope between 30 and 60 minutes is smaller than the slope between 1 and 15 minutes, again it sounds like you're seeing a chamber artifact where concentrations in the chamber are approaching equilibrium with concentrations in the soil, and you might be better o9 using data from 1 to 30 minutes.

Response: Using the coefficient of the non-quadratic term in a second order polynomial is equivalent to estimating the flux from the initial slope of the curve

Line 249: Instead of 'scaling up' I think it would be more precise to say that you're estimating a flux that is representative of the entire plot, and it's being done by calculating a weighted mean of fluxes from the basin and interrow chambers. You could write something like "We estimated a flux representative flux for each plot ($N_2O_{plot}$) by calculating a weighted mean of fluxes from the basin and interrow chambers. We used weighting factors of 0.12 for basin . . .".

Response: Thank you for the suggestion, we have now rephrased as follows: We estimated a representative flux for each plot (area-weighted cumulative $N_2O$ emission) by calculating a weighted mean of fluxes from the basin and interrow positions. Weighing factors of 0.12 and 0.88 were used for basin and interrow areas,

respectively, in CA treatments (CA and CA+BC), while a factor of 0.50 was applied to both inrow and interrow areas in conventional treatments (Conventional and ConventMM). (Ln 248 – 252).

Line 259: indicate when extractions were conducted relative to sampling (e.g., x hours, the next day, etc)

Response: We have now added the following sentence: The soil samples were extracted the same day, within 5 hours after sampling (Ln 260).

Line 280: I would change "dividing the scaled cumulative N2O emissions . . ." to something like "dividing the weighted mean N2O emissions" or "dividing $N_2O_{plot}$ emissions. . ."

Response: Since we earlier mentioned how scaled cumulative $N_2O$ emissions were calculated (Ln 248 - 258), we have rephrased the sentence as follows: Yield-scaled $N_2O$ emissions (kg $N_2O$-N kg$^{-1}$ grain yield) and N-yield scaled emissions (kg $N_2O$-N kg$^{-1}$ grain N) were estimated for each season by dividing the area-weighted cumulative $N_2O$ emissions with grain yield or N content of the grain (N concentration × grain yield).

Line 281: I would change "scaling factor" to "weight"
Response: done

Line 283: I would change "scaling factor" to "weight"
Response: done

line 293, you indicate that random e9ects were introduced to account for repeated measurements. I need more justification for this: typically, one assumes that repeated measures of the same plot will be correlated to some extent. And since statistics are only conducted on cumulative variables (which is perfectly reasonable, and a good way to avoid issues of autocorrelation in repeated measures), I'm not sure why time is discussed as an e9ect at all.

Response: Repeated measurements (frequency) was used as a random effect on hourly N2O fluxes and mineral N, not cumulative N2O (check Table S1 and S2 in the supplementary material). N2O flux data was used for regression analyses with WFPS and mineral N.

Line 296-298: I find it rather remarkable that no data transformations were required (I've never seen field data like this that didn't).

Response: For cumulative $N_2O$ emissions, we checked the normality of residuals. Visual inspection of the QQ plots showed that residuals were near normal without transformation as shown in fig 1.

[Figure]

Fig 1: Cumulative $N_2O$

For N2O fluxes, The QQ plot shows deviation from normality in the tails, which is common for $N_2O$ flux data. However, mixed-effects models are robust to violations of normality in residuals (Zuur et al., 2009), especially with large sample sizes. We tested log transformation, but it did not improve model fit (AIC/BIC remained similar), so we retained the original scale for interpretability.

[Figure]

Fig 2: N$_2$O fluxes

[Figure]

Fig 3: Original N$_2$O fluxes vs log transformed N$_2$O fluxes

We have rephrased and included this information in the manuscript as follows: We validated model assumptions by checking quantile plots of residuals against fitted values. Visual inspection of QQ plots showed that residuals were approximately normally distributed for cumulative N$_2$O data. However, N$_2$O flux data were transformed and did not substantially improve model fit, so we retained the original scale of interpretability. Mixed-effects models are robust to mild non-normality (Zuur et al., 2009). (Ln 295 - 299)

Line 333: change "were" to "when"

Response: done

Line 349: change "second respectively" to "second season, respectively"

Response: done, thank you.

Figure 2 & 3: I think you want to combine these into a single figure—I don't think it's necessary to compare Conv to convMM in one figure and then compare Conv to the CA treatments in a separate figure, especially since you're presenting post-hoc contrasts.

Response: In this study we tested the effect of (1) rotation and (2) tillage. This is stated in Ln 112 – 115 as follows: Specifically, we compared crop rotation (pigeon pea – maize) with maize monocropping under conventional tillage (ConventMM). In addition, we compared pigeon pea – maize rotation under three practices, i.e. conventional tillage, CA (reduced tillage), and CA in combination with biochar (CA+BC).

Therefore, we think it is relevant to keep Fig 2 and 3 as separate. Fig 2 is comparing the effect of rotation under conventional tillage and Fig 3 is comparing effect of different tillage practices and biochar on mineral N.

The higher nH4+ in convMM than all other treatments in season 1 seems unexpected, and (maybe) higher or equivalent to the CA pigeon pea treatments in both years? Something that may be worth discussing.

Response: Thank you for the comment, we have now included this in the discussion as follows:

Generally, NH4+ and NO3- contents were more variable in the first season (May – October) than the second season (October – January) (Fig 1b, 1c, 3). At the onset of the experiment, mineral N was most likely from mineralisation of chemically mulched grasses having grown on the fallow for 3 years prior to the experiment. Higher NH4+ in maize than pigeon pea in the first season under conventional tillage might be attributed to differences in crop phenology. Pigeon pea is a slow starter; its nodulation and peak biological N2-fixation typically occur around 80 days after sowing. Therefore, during early establishment phase, April – June, pigeon pea relies on NH4+. Legumes generally show a stronger affinity for $NH_4^+$ in their early growth stages due to lower energy cost for its assimilation. Consequently, efficient soil N uptake by pigeon pea in the early stages reduces soil $NH_4^+$ levels compared to maize, which generally exhibits slower early-season N uptake.  (Ln 484 – 494).

Line 388: Delete the text "Fig. 4" -- it does not present cumulative emissions

Response: deleted

Line 392: Interpreting the treatment main effect in the presence of a significant treatment x position interaction (which says that the treatment effect depends on the chamber position) is complicated—another reason to use the weighted mean flux for each plot as the response variable. But I do believe interpretation of a significant main e9ect in the presence of an interaction in an RCB design does indicate a main e9ect that is over and above the treatment x block interaction.

Response: We agree, in addition to the position effect, we have added the weighted N2O emissions.

Figures 5 and 6: I think you want to combine these into a single figure—I don't think it's necessary to compare Conv to convMM in one figure and then compare Conv to the CA treatments in a separate figure, especially since you're presenting post-hoc contrasts.

Response: In this study we tested the effect of (1) rotation and (2) tillage. This is stated in Ln 112 – 115 as follows: Specifically, we compared crop rotation (pigeon pea – maize) with maize monocropping under conventional tillage (ConventMM). In addition, we compared pigeon pea – maize rotation under three practices, i.e. conventional tillage, CA (reduced tillage), and CA in combination with biochar (CA+BC).

Therefore, we think it is relevant to keep Fig 2 and 3 as separate. Fig 2 is comparing the effect of rotation under conventional tillage and Fig 3 is comparing effect of different tillage practices and biochar on N2O emissions.

Table 3:

- It would be helpful to have a reminder in the Table caption that the weighted mean fluxes were used to calculate yield-scaled emissions.
- I would add something to the discussion explaining why we would care about Nyield scaled emissions.
- Why are the season comparisons (capital letters) only included for ConventMM and not the other treatments?

- I would explain in the table caption why ConventMM is not included in the treatment tests (lower case letters) in Season 1 (presumably because comparing maize yields to pigeon pea yield isn't a useful comparison).

- I think it would be useful to include the P values rather than "ns." It provides the reader with more information and context for interpreting the results. A P value of
0.08 is very different from a P value of 0.5

Response: We have now updated Table 3 caption as follows: Grain yield, grain N yield, weighted N2O, yield scaled N2O emissions and N yield scaled N2O emissions during the first and second rain season, in northern Uganda. When calculating yield-scaled N2O emissions, a weighing factor of 0.12 in planting basins and 0.88 in interrows was used in CA treatments (CA and CA+BC), and a weighing factor of 0.50 was used for both inrows and interrows in conventional treatments (ConventMM and Conventional). Means are shown with standard errors of means (N=4). Uppercase letters compare seasons specifically for a monocrop treatment (ConventMM), maize was grown in both seasons. Lowercase letters compare treatments with rotation (Conventional, CA, CA+BC) within a season. Different letters represent significant differences (p < 0.05), determined at 5% level using Tukey test, ns represents not significant (p > 0.05). (Ln 416 – 425).

We have now added N yield emissions in the discussion.

Line 434-443: I would just make it explicit here that you are not including ConvMM in treatment comparisons for season 1 since maize was grown in that treatment rather than pigeon pea, as in all the other treatments.

Response: done

Line 442: "significantly higher" needs to be changed: Maize yield was higher, but yieldscaled emissions were lower

Response: Thank you, we have corrected the phrase, and it now reads as follows: Maize yield in ConventMM were significantly higher (p < 0.05) in the second season compared to the first season, while yield scaled emissions were greater in the first season.

 I think it would be useful to compare emissions to conventional agricultural settings as well—control plots from other experiments in East Africa. It may also be useful to provide emissions—including yield-scaled emissions--from conventional management using fertilizer as context for the conventional and CA plots in this study, while also discussing issues associated with no-input agriculture such as nutrient depletion—how long could these different practices remain sustainable?

Response: Thank you for the comment, we have now compared our results with the conventional treatments in East Africa where fertilizer was not applied. Our main challenge is finding adequate literature with studies where inorganic fertilizers were not applied, we only got two papers and we have included them in the discussion as follows: Area-weighted cumulative N2O emission in the conventional treatments ranged 0.3 - 0.6 N2O ha-1 in 1st and 2nd season, respectively. These results are consistent with Baggs et al., 2006 and Bwana et al., 2021 who also recorded $N_2O$ emissions of 0.2 and 0.5 kg $N_2O$ ha$^{-1}$ in conventional treatments were inorganic fertilizer was not applied (Ln 513 – 516).

We have now included this paragraph: A key challenge for the sustainability of unfertilized agroecosystems is the management of soil nutrient balances. While biochar amendments in CA systems can effectively reduce $N_2O$ emissions and maintain crop productivity, these systems gradually deplete soil nutrient reserves. Although $N_2O$ is a powerful GHG, it represents a relatively small component of the total annual N budget, often less than 1 kg N ha$^{-1}$. The primary source of nutrient removal is the export of grain, which removes a significant amount of N from the system. While biological $N_2$-fixation from pigeon pea can fix up to 110 kg N ha$^{-1}$, some of this input can as well be exported from the field in the stalks, and the remaining N in the form of decaying roots, rhizodeposits and leaf fall is often insufficient to fully compensate for the N removed by the exported grain. Furthermore, the biologically fixed N is prone to rapid mineralization and subsequent loss through leaching or other pathways before the next crop can fully utilize it. To achieve long term sustainability, an intergrated approach is required, including targeted fertilization to prevent continuous nutrient mining and to ensure the long-term viability of the agroecosystem. (Ln 577 – 590).

 https://doi.org/10.1029/2020JG005742 could be helpful for the rewetting discussion.

Response: We really appreciate the provided link. We have now rephrased as follows: A similar, though smaller emission peak was observed in June when abundant rainfall terminated a dry spell. Rewetting of dry soil triggers $N_2O$ emissions likely due to increased nitrification and denitrification fueled by release of readily

available N and C from dead microbial biomass (Namoi et al., 2019). Birch Effect is an important source of $N_2O$ emissions in seasonally dry ecosystems (Hickman et al., 2020) (Ln 465 - 468).

Line 487: much of the fixed N may be in the harvested biomass, limiting the amount returned to soil.

Response: We have now included this in our discussion, please check Ln 574 – 587.

Line 494: I don't think NH4 is higher in CA systems than conventional systems—isn't the NH4 in ConvMM no di9erent than the CA treatments? And if ConvMM is higher than conv, it doesn't follow that the mineralization of pigeon pea residues is responsible for the di9erence. It also conflicts with the statement in line 486.

Response: Thank you for the comment, we have now deleted the sentence.

Line 514-516: this sentence is missing something, or 'nutrient cycling' should be deleted.

Response:      nutrient cycling now deleted

Line 540-542: You may want to rephrase this. Ff no treatment e9ects were expected, it would seem to undermine a justification for the entire experiment (i.e., readers may ask why measurements were conducted in a year when no treatment e9ect was expected, instead of at a time when the treatments would be expected to have an e9ect).

Response: Thank you, we have now deleted the statement.

Line 566: delete "We established that"

Response: deleted

---

## Author Comment (AC3)

**Reviewer 1**

Response to review of the manuscript "Nitrous oxide emission from pigeon pea – maize rotation in response to conservation agriculture and biochar amendments in a Ferralsol, northern Uganda" by Namatsheve et al.

This study investigates the effects of conservation agriculture (CA), both with and without biochar amendments, compared to conventional tillage in an unfertilised maize-pigeon pea rotation in Uganda. It also compares these treatments with a maize monoculture treatment.

The manuscript significantly contributes to the scientific knowledge base by presenting a novel dataset.

The introduction shows the relevance of the study and the basis of existing literature. The study design and hypotheses are well described. The data is highly valuable. It contains 17 observations of $N_2O$ emissions and its driver variables per treatment over the course of one year (two seasons) in Uganda. The methods are sound and well described. The results are well described and the discussion a good start, but both need some revisions as suggested below.

I recommend to publish the manuscript after major revisions.

Response: We thank the reviewer for the constructive evaluation of our manuscript. We appreciate the recognition of the novelty and relevance of our dataset, the clarity of our study design and hypotheses, and the value of our methodological approach.

We have carefully addressed all specific comments and suggestions to improve the clarity and scientific rigor of the results and discussion sections. We believe these revisions have strengthened the manuscript and hope it now meets the standards for publication in Biogeosciences. All line numbers given in the responses refer to the R1 version of the manuscript.

L30: Overall, across all seasons, cumulative growing-season (279 days) N2O emissions ranged from 0.44 – 1.11 kg N ha-1

RC: Since this is a major finding, you should give the values specifically per treatment in the abstract.

Response: We agree with the reviewer and have now added the overall values to the abstract (Ln 31 - 33): "Overall, across two growing seasons, area-weighted cumulative $N_2O$ emissions ranged from 0.46 kg in CA+BC treatment to 0.88 kg N ha$^{-1}$ 279 days$^{-1}$ in the Conventional treatment, respectively."

RC: You start with Table S19. You should instead name your Tables differently and make sure to have them ordered in the same way as they occur in your text. You start with table S1 here and so on.

Response: Thank you for the comment, we have now rearranged the tables, and numbered them in logical order (viz. table S1).

L130: RC: Please add average precipitation sums (for example 5 or 10 years) in order to show how usual the 2023 conditions were.

Response: We have added following information (Ln 132–135): *Mean annual rainfall in the period 1981 – 2010 in Gulu was 1460 mm yr$^{-1}$ (Oriangi et al., 2024). The annual rainfall in 2023 was 1238 mm, of which 818 mm precipitated in the first and 419 mm in the second season, respectively. Average temperature for 2023 was 24 ℃.* .

L135: Is it known how long the maize and cassava production have been practiced?

Response: We now include following information (Ln 138–139): *The experiment was established on a field that was fallowed for three years; before that it was used for maize and cassava production without fertilization, for at least five years*.

L160: For CA, weeds were controlled by spraying glyphosate at a rate of 1.03 L ha-1, immediately after sowing and hand pulling throughout the season. RC: Set in context how realistic glyphosate spraying would be under farmers practice conditions.

Response: Subsistence farmers normally control weeds by hand hoeing. Conservation agriculturalists promote herbicide application to avoid tillage and to control troublesome perennial weeds that are difficult to control by hand hoeing. Although the use of glyphosate among farmers is a relatively new technology, it is readily available at agro-dealers within the smallholder farming communities, and some farmers use it.

L 198: RC Write: Plot wise sampling in 0 – 20 cm was carried out at the onset (April 2023) and end (October 2023) of the first growing season from planting basins in CA and CA+Biochar treatments and in the planting rows in conventional treatments to assess the effect of different treatments on soil properties. Leave out the next sentence.

Response: Thank you for rephrasing L198. We have now deleted the next sentence

L234: by (Žurovec et al., 2017). RC: write by Žurovec et al. (2017).

Response: Thank you for the comment, the reference is now correctly cited.

L246: $\sum (fi + (fi+1)/2 \times (ti+1 - ti) \times 24 \times 10-5$. RC: There is a bug in this formula, there are opening up three brackets but only two of them close again. Make sure to provide a correct and well formatted equation.

Response: Thank you for the comment, the formulae is now written as follows:

$$cumulative\ N_2O = \sum[(f_i + f_{i+1})/2 \times (t_{i+1} - t_i) \times 24 \times 10^{-5}] \qquad (2)$$

L255: (NO3-N and $NH_4^+$-N). RC: Use $NO_3^-$ here and elsewhere (Text & Fig. to ensure consistency.

Response: We have changed this in the whole manuscript.

L281-283: Adjust the text, since it is the same as before. RC: My suggestion: For calculating yield scaled N2O emissions, the same scaling factor as for cumulative N2O fluxes of 0.12 for basin and 0.88 for interrows in CA treatments, and 0.50 for inrows and interrows in conventional treatments were applied.

Response: We have rephrased the sentence as follows (Ln 281 - 284): *Yield-scaled $N_2O$ emissions (kg $N_2O$-N $kg^1$ grain yield) and N-yield scaled emissions (kg $N_2O$-N $kg^{-1}$ grain N) were estimated for each season by dividing the area-weighted cumulative $N_2O$ emissions with grain yield or N content of the grain (N concentration × grain yield).*

L297: RC: The statistical approach makes sense. Nevertheless, I wonder how residuals data can be normal without any transformation, this is very unusual for N2O, could you show the residual plot?

Response: For cumulative $N_2O$ emissions, we checked the normality of residuals. Visual inspection of the QQ plots showed that residuals were near normally distributed (Fig. 1), hence the data did not require any transformation.

[Figure]

Fig 1: Residual plots for cumulative $N_2O$ emissions

For hourly N2O fluxes, residual plots showed a deviation from normality, which is common for $N_2O$ flux data. However, mixed-effects models are robust to violations of normality in residuals (Zuur et al., 2009), especially with large sample sizes. We tested log transformation, but this did not improve model fit (AIC/BIC remained similar), so we retained the original scale for interpretability.

[Figure]

Fig 2: Hourly $N_2O$ fluxes

[Figure]

Fig 3: Q-Q plots comparing non-transformed and log transformed hourly $N_2O$ fluxes

L310 3.0 Result -> 3 Results

Response: Done

L316: SOC ranged from 1.25 – 2.23% and biochar significantly increased C. RC: use SOC consistently since this is the abbreviation you introduced.

Response: Thank you for noting this;  we now use SOC throughout the manuscript.

Figure 1: The insert in (a) shows mean ± se N2O fluxes during peak emission on 18 October 2023.

RC: This is the same as you see when looking at this peak day in the time series plot, so I do not see the additional value to much. Well if you want to pronounce it a lot you can keep it. I would rather show the cumulative values - but this might show up later in a separate Figure?

Response: we removed the insert in (a)

Table 3: Please round the yields meaningfully (Consider the accuracy of measurements!).  Give no decimal number here if you have kg /ha. Same for N yields.

Response: Thank you for the comment; we have rounded off the grain and N yields to whole numbers.

Figure 5+6: Add the **weighted cumulative** values to the plots as a third colour besides basin/inrow and interrow, since this is the main result that should be also show here. Indicate Tukey-HSD letters also in Fig 5

Response: Thank you for the comment. We now include weighted cumulative values in figures 4 and 5. We also added letters to show significant differences in figure 5 (Ln 397 – 413).

L388: Write two sentences, else it is confusing: Chamber position did not significantly explain variations in cumulative N2O emissions. There were no significant differences ($p > 0.05$) between conventionally tilled pigeon pea – maize rotation and conventionally tilled continuous maize monocropping (Table S17, S18, Fig. 5, Fig S3)

Response: Corrected

L477: corelated -> correlated: Please check spelling (and grammar) throughout the manuscript- I didn't.

Response: corrected

L481: probably immobilizing available N rather than releasing it for microbial N transformations; RC: I know what you mean but… The phrase "releasing it for microbial N transformations" is a bit unclear — microbes don't exactly "release" N for other microbes; it may be better to phrase it differently:

Better: This suggests that microbes in these soils compete effectively for mineral nitrogen, likely immobilizing it and thereby reducing its availability for microbial processes such as nitrification and denitrification

Response: Thank you for this helpful suggestion. The sentence is now rephrased as follows (Ln 478 - 480): *This suggest that microbes in these soils compete effectively for mineral N, likely immobilizing N and thereby reducing its availability for microbial N transformations such as nitrification and denitrification*.

L485: Our results imply that the process of symbiotic N fixation per se, and residue retention do not affect soil mineral N and N2O emissions in unfertilized soils with inherently low N. Rochette et al. (2004) also reported that there is considerable uncertainty related to the emissions of N2O from soils under legumes, and the soil mineral N alone was a poor indicator of N2O emissions for two seasons in acidic soils in Canada. RC: 1) It is well-known that N2 fixation itself does not affect N2O emissions, please rephrase so this gets clear. 2) In many studies soil mineral N is a bad indicator for N2O emissions. This is nothing particular to legumes. Please adjust.

Response: We thank the reviewer for the insightful comments. We agree that there is no direct link between $N_2$-fixation and $N_2O$ emissions. We also acknowledge that soil mineral N is a poor indicator of $N_2O$ emissions. We have now rephrased as follows (Ln 484 - 488): *Our results imply that $N_2$-fixation and residue retention do not directly affect soil mineral N or $N_2O$ emissions in unfertilized soils with inherently low N. Rochette et al. (2004) also reported considerable uncertainty in $N_2O$ emission from soils under legumes, and they noted that soil mineral N alone was a poor predictor of $N_2O$ emissions for two seasons in acidic soils in Canada*.

L536: Lentz et al. (2014) also found that biochar reduced N2O emissions by 50%, indicating that biochar inhibited nitrification and N immobilisation. RC: Did they measure nitrification and immobilisation? It is a bit contrary to the study by Munera-Echeverri et al. (2022), so the effect of biochar on nitrification and the difference between these studies should be explained.

Response: Lentz et al., 2014 measured $N_2O$ emissions and net mineralization, and attributed differences to nitrification and N immobilisation. We agree that Lentz et al. (2014) contradict Munera-Echeverri et al. (2022). We have rephrased the whole paragraph, as we now discuss the area-weighted N2O emissions results.

The entire paragraph now reads (Ln 539 – 556): *Biochar applied at a rate of 5 Mg ha$^{-1}$ to CA systems reduced area-weighted $N_2O$ emissions by 33% and 66% compared to Conventional treatment in the first and second season, respectively. Biochar with a high C:N ratio of >60, as applied in this study, was shown to reduce the bioavailability of inorganic N through microbial immobilisation (Namoi et al., 2019) or sorption of $NO_3^-$ due to unconventional H-bonding between $NO_3^-$ ions and biochar surface functional groups (Kammann et al., 2017; Nguyen et al., 2017). In a fertilized Ultisol in western Kenya, Fungo et al. (2019) reported a 22% reduction in emissions. Case et al. (2015) also reported that biochar suppressed $N_2O$ emissions in a sandy*

*loam soil fertilized with 140 kg N ha$^{-1}$ yr$^{-1}$. Biochar applied in a calcareous soil also reduced N$_2$O emissions by 50%, and increased soil NH$_4^+$/NO$_3^-$ ratio, indicating that biochar impaired nitrification and N immobilisation processes (Lentz et al., 2014). Our results support the growing evidence that biochar can mitigate N$_2$O emissions via various N cycling modifications (Liu et al 2018, Zhang 2021, Borchard et al., 2018; Kammann et al., 2017). However, in a short-term study, Munera-Echeverri et al. (2022) reported that biochar amendments did not affect N$_2$O emissions in Zambia despite increased gross nitrification rates in the biochar treatments. In a global meta-analysis, Shakoor et al. (2021) showed that biochar increased N$_2$O emissions by 20%. Discrepancies on response of N$_2$O emissions to biochar might be explained by biochar type, soil parameters, climatic conditions and experimental duration.*

L554–561: End the paragraph with your study, not with the numbers of another study. Put the following sentence to line 554: Our results are in line with other studies in SSA, although they applied mineral N fertilizer; for example, Shumba et al. (2023) reported yield scaled emissions of 0.09 – 0.19 in maize after applying 58 kg N ha-1 in a Ferralsol and Lixisol in Zimbabwe. Rephrase to make it consistent again. Then end the discussion with this: These practices were effective in minimising emissions without penalising pigeon pea productivity, supporting CA as a sustainable agricultural practice.

Response: Thank you for the useful suggestions which we have taken into account when rephrasing the paragraph which now reads (Ln 573 – 577). *Our results are in line with other studies in SSA, although they applied mineral N fertilizer; for example, Shumba et al. (2023) reported yield scaled emissions of 0.09 – 0.19 in maize after applying 58 kg N ha$^{-1}$ in a Ferralsol and Lixisol in Zimbabwe. These practices were effective in minimising emissions without penalising pigeon pea yields. This supports CA as a sustainable agricultural practice.*

L569: Typo: Yield-scaled

Response: corrected

---

## Author Comment (AC4)

**Reviewer 2**

The authors have conducted an experiment in Gulu, Uganda in which the e9ects of various conventional and conservation agricultural treatments on crop productivity, soil N dynamics, and N2O emissions are analyzed. The study is conducted over two cropping seasons within a single year – April – October 2023 and October-Jan 2024. Plots that had been fallow for the previous 3 years were cleared and prepared in an RCB design with four treatments: conventional maize monocrop (ConMM), conventional pigeon pea/maize (Con) rotation, conservation agriculture (CA), and conservation agriculture with biochar amendments (CA+bc). Some di9erences were observed in NH4 but not in NO3. Maize grain yield in season 1 was lower in conMM than pigeon pea grain yield in the other treatments; there was no di9erence in grain yields in the second season. Yield-scaled N2O emissions were higher from the conventional than the CA treatments in the first season, but there were no di9erences in the second season.

This study adds much-needed measurements of N cycling and N2O emissions from low/no-input cropping systems in Uganda, and I look forward to seeing them in the peer reviewed literature. I do have several general comments.

Major comments:

A major limitation of this study is that data are only presented from a single year (which encompasses a long-rain and a short-rain cropping season). Interannual variability can play a big role in crop productivity and N cycling (highlighted by the statement in lines 548550 that drought drastically reduced maize grain yield in the first season), and this type of study would benefit from more than one year of measurements. It is true that two cropping seasons are included, but these are di9erent cropping seasons – long rains and short rains—and cannot be said to represent the same thing as multiple years of measurements. The fact that there are only one year of data presented here and, as a result, it is not really possible to generalize really needs to be emphasized front-and-center in the text. (There are so many examples in ecological studies of a second year of data drastically changing results that it's become cliché to joke that "I should have quite collecting data after one year!") That said, 1 year of data can still be quite useful, and I think it's particularly important to have these data published because there are so few data from the region, and particularly few data from pigeon pea systems. And I am especially sympathetic to the tremendous amount of e9ort required to establish these experimental plots and to collect and analyze these data. I leave it to the editor to decide whether a revised manuscript would meet biogeosciences' editorial threshold for publication.

Response: We thank the reviewer for the positive appraisal and agree that data from several subsequent seasons would have been valuable. However, due to budget and time constraints, this was not feasible. Nevertheless, we believe that presenting effects of different management practices on $N_2O$ emissions from an unfertilized system in SSA for two growing seasons is of interest to both the scientific community and stakeholders. Since the effects of biochar on soil chemical properties are expected to diminish over time (e.g., Cornelissen et al. 2018), measuring emissions immediately after biochar application for one full year provides some insight.

On a somewhat related note, the authors' suggestion that treatment differences were not expected in this first year (e.g., Line 540-542) would seem to raise a fundamental question: if treatment differences were not expected, then why were measurements conducted in the first year, and why is it interesting to publish them? A way to address this concern would be to cite examples from the literature where treatment differences (when similar treatments were compared) *were* found in the first year of establishment.

Response: We agree that the sentence about expected effects did invalidate our study to some extend. We have therefore removed the sentence. We have now cited other studies with treatments effects in the first year.

**Other general comments:**

I don't understand why chamber position is included as a treatment in statistical analyses. If there were some targeted analyses into mechanism then it might make sense, but in this manuscript the central questions are how the agricultural management treatments differ, and how soils change over time within an agricultural management treatment. To answer this question, the weighted mean of the within row and interrow chambers would be the appropriate response variable— i.e., an estimate of the flux from each plot.

Response: Chamber position was included as a factor in the statistical analysis to assess potential differences between basins/inrow where maize or pigeon pea was planted) vs. interrow (no planting, no soil disturbance under CA, but overall digging under conventional treatment). A distinction between basin and inrow was also needed to discern biochar effects, as biochar was only added to the basins and not to inrow soil. Abiven et al. (2015) found root biomass inside planting basins to be twice as large in biochar amended plots compared to regular CA in Zambia. In this unfertilized system, the main difference between basins and interrow location is that basins are re-opened every season under CA and that biochar is added to the basins only. Effects of biochar addition on soil properties were expected in the basins where biochar was applied, not the interrows.

In addition to reporting differences between basins/inrow and interrow positions, we now include area-weighted cumulative N2O emissions in Fig 4 and 5, as well as yield- and yield-N scaled cumulative N2O emissions (Table 3) to show estimated total fluxes for the different management practices, area-weighted for basins/inrow and interrow positions.

I think there's a missed opportunity for more discussion of what makes this study unique-the focus on unfertilized systems. Currently this is not discussed much at all. There are actually quite a lot of N2O data from unfertilized maize in tropical African systems that could provide context for understanding the emissions in this study– the control plots from fertilizer trials. Off the top of my head, there are studies in East Africa with lead authors that include Baggs, Hickman, Millar, Pelster, Rosenstock, Tully, Zheng, and studies in Zimbabwe from Mapanda. Typically there are multiple studies from each author, and at least some of them include explicit calculations of yield-scaled emissions.

Response: We agree with the reviewer that the uniqueness of our study is the absence of fertilization, next to the use of legumes and biochar for CA. However, based on a recent review of 26 published studies dealing specifically with N2O emissions in SSA (Namatsheve et al., 2024; see figure below), at least 20 kg N ha$^{-1}$ was applied.

[Figure]

*Write a figure caption here*

We now include this information in the manuscript as follows (Ln 104 – 106): *In our recent meta-analysis of CA and biochar effects on N cycling, we found that residue retention increased soil $NO_3^-$, leading to higher $N_2O$ emissions; this finding was based on 26 published studies focusing specifically on $N_2O$ emissions, and applying at least 23 kg N ha $^{-1}$.*

Initial soil sampling for NO3- and NH4 analyses was conducted after treatment application.  Since there were no NO3- and NH4+ measurements before treatment, these measurements are able to answer one question: how treatment affects changes in NO3- and NH4+ snap shots when comparing the start of the first crop season to the end of the second crop season of a given year. I think it would be very useful to include some discussion/contextualization as to why this is an important question—it's not immediately clear to me.

Response: We do not have the background values of mineral N from before the experiment. The absolute amounts especially for the first season were mainly influenced by the mineralisation of organic matter, built up during the 3-year fallow period before establishing the experiment. The amounts of NH4+ and NO3- in the soil were investigated as potentially important explanatory variables for N2O

emission and to explore the effect of different soil managements in soil mineral N. Relative differences should be valis.

**Specific comments:**

Line 40: delete "tropospheric"

Response: deleted

Line 42: N2O is actually the largest anthropogenic driver of stratospheric ozone depletion (see Ravishankara 2009 in Science DOI: 10.1126/science.1176985)
Response: Thank you for the comment, we have now included the provided reference.

Line 44: replace 'atmospheric' with 'anthropogenic' and change to "60% to global **anthropogenic** N2O emissions"

Response: done

Line 60: you can delete 'or by-product'

Response: done; however, in nitrification, N2O is a by-product, not an intermediate.

Line 64: also labile carbon availability ('substrate availability' would most likely be interpreted to refer to nitrogen compounds)

Response: We thank the reviewer fir this comment and have changed the sentence accordingly, which now reads (Ln xxx-yyy): *The biogeochemistry of $N_2O$ in soil is to a large extent regulated by complex interactions between environmental and biogeochemical factors such as temperature, water, labile carbon availability, oxygen levels, acidity and substrate availability (Case et al., 2015; Tian et al., 2020).*

Line 137: indicate in the text when specifically the clearing was conducted and when glyphosate was applied.

Response: We now give this information in (Ln 138 – 141): *Prior to the establishment of the experiment, on 15 March 2023, a dense, naturally grown vegetation of grasses and shrubs was removed by slashing and chemical weeding using glyphosate [N-(phosphonomethyl) glycine].*

Line 138: I'm not an expert in herbicides, but I think glyphosate can potentially have a range of impacts on crop production (via uptake by crops either from glyphosate that has persisted in soil or is released from decomposing weed biomass; it also has potential interactions with mineral nutrition: see https://pmc.ncbi.nlm.nih.gov/articles/PMC6918143/), which makes the timing of its use useful to know, and which might have implications for season 1 production.

Response: Thank the reviewer for this comment and appreciate the reference to relevant literature. In our study, glyphosphate was applied on 15 March 2023, 19 days prior to sowing in the season 1 (4 May). We agree that the short period between spraying and sowing might not have been enough to fully degrade the chemical. The glyphosate remaining in soil may therefore have had potential impacts on crop production and mineral nutrition, but there was no possibility to determine its residual concentration. Given the high temperatures, however, we believe that residual glyphosate concencentrations decayed quickly. Despite these potential challenges, glyphosate is promoted by the mainstream of the conservation agriculture community where controlling perennial weeds with mechanical and biological methods is a challenge, especially in early years of adoption.

Line 139: indicate in the text specifically when the plots were tilled

Response: Date for tillage is now given in Ln 142 – 145: *On 27 March 2023, plots under conventional management were prepared by overall digging using hand hoeing (100% tillage) and plots of the same size under CA by manually digging 10-L planting basins (35cm long × 15cm wide × 20cm deep) spaced 70 cm × 35 cm (interrow × within row spacing).*

Line 144: ConventMM has not been defined at this point. The CA+BC abbreviation has also not been introduced—it was introduced as CA+biochar.

Response: Thank you for this comment. We have now consistently changed 'CA+biochar' to 'CA+BC' throughout the manuscript, and introduce 'ConventMM' in abstract and introduction.

Line 157-158: I'm a little confused by plant spacing. In line 141, it sounded like the spacing was 70cm x 35 cm, which would yield a planting density of 40,898. Here it says 10cm; if that's 10cm x 70cm, that would be a planting density of 143,143. And were the same planting densities really used for maize and pigeon pea?

Response: Plant densities for both maize and pigeon pea were the same. We accounted for the size of the basins which were 35 cm long, 15 cm wide and 20 cm deep. Therefore, if the spacing for the basins is 70 x 35 cm. With 3 seeds in each basin, one ha has 21 000 basins and a planting population of 63 000 plants ha$^{-1}$.

Line 175-176: two different biochar treatments are described here, including one that isn't defined or mentioned anywhere else in the manuscript (CA+BC+BC).

Response: Thank you for noting this. We have now deleted CA+BC+BC, as it is not part of this paper.

Line 213: Be specific about when the collars were installed (you can say something like 'at least xx days before the first measurement).

Response: Thank you for this comment. We have rephrased the sentence as follows (Ln 203 - 205): *Permanent gas sampling plots were established by inserting 17 cm diameter PVC rings (the base) to a depth of 7 cm into the soil on 19 April 2023, 3 weeks before first sampling on 10 May 2023*.

Line 210-211: I don't think I quite have a complete description of the chamber top— it sounds like it should be a pvc pipe – is it a pipe that was manufactured with a top, or did you have to add and seal a top to the pipe?

Response: We used prefabricated lids as top on sewer pipe lid (figure below and Fig. S2). On top of the lids, we drilled holes and fitted it to a self-sealing rubber septum for gas sampling.

[Figure]

[Figure]

Line 218: citing a reference here for using petroleum jelly would be great. It sounds like it would work, but a reference would be helpful.

Response: We have included a citation and the sentence now reads (Ln 209 - 210) : *To facilitate chamber deployment, the contact area between the collar and chamber was sealed with a thin layer of petroleum jelly as described by Shumba et al., (2023)*.

Line 220: were the initial measurements really taken at 1 minute? (i.e. sampling was delayed until the chamber had been sealed for a full 60 seconds)

Response: Yes, the first sample was taken during the first minute; in addition to sealing the chamber, the plunger of the syringe was pumped three times in and out before a representative sample was obtained. This took 0.5 – 1 minute.

Line 220: 60 minutes is a fairly long period. Can you establish that N2O did not reach or approach equilibrium in the chamber?

Response: We measured concentration in the chambers after 1, 15, 30 and 60 minutes. The 60-minute period was chosen because we anticipated relatively low $N_2O$ emission rates and also determined CH4 uptake (which will be reported in another study). In some cases, chamber N2O kinetics indicated that equilibrium was approached, i.e. concentration raise slowed down. In these cases we applied quadratic fits to emphasize the "time-zero" rate as given by the coefficient of the

non-quadratic term, which comes close to estimating the flux from the first two measurement points.

We established that in most cases N2O did not reach equilibrium in the chamber. This is stated in the manuscript (Ln 230 – 231) as follows: *Plotting measured $CO_2$ and $N_2O$ concentrations over time, revealed linear increase in most cases with little saturation observed*. In some cases, the $N_2O$ concentration in the sample taken right after chamber deployment was substantially higher than 0.336 ppm (ambient $N_2O$ concentration), pointing at residual $N_2O$ in the chamber. The exalted $N_2O$ concentration after chamber deployment usually decreased until the second measurements (15 min) and to avoid fitting negative fluxes, the first sampling point was discarded. *Flux rates were estimated by fitting a linear or second order (polynomial) function to the concentration change over time. A quadratic fit was only used in few cases in which $N_2O$ accumulation in the chamber showed a convex downwards trend, i.e., decreasing emissions* (Ln 230 – 238).

Line 228: Did you have any checks to make sure there wasn't any sample contamination/leakage during shipping? (e.g., having vials filled with a standard gas in Uganda and then measured in Norway.)

Response: He-filled vials were included as blanks to check for contamination during storage and shipment of the vials. Detected concentrations of $CO_2$ and $N_2O$ were <5% of ambient. We used slightly over-pressured, crimp sealed glass vials with thick butyl septa, which have been shown previously to maintain pressure and mixing ratios during air transport and storage (Raji and Dörsch, 2020). (Ln 220 - 224).

Line 235: how was it determined when a linear or when a polynomial fit should be used? (i.e., it would be good to make sure overfitting was avoided; the additional parameter in the polynomial model can provide a closer fit to the data even if it is inferior to the linear model)

Response: see Ln 235 – 238: *Flux rates were estimated by fitting a linear or second order (polynomial) function to the concentration change over time. A quadratic fit was only used in few cases in which $N_2O$ accumulation in the chamber showed a convex downwards trend, i.e., decreasing emissions*.

Line 242: a quadratic fit does not seem like an appropriate way to calculate a flux here, and makes me a bit worried about the polynomial fits in general. If concentrations are convex downward, that suggests to me that concentrations in the chamber have equilibrated with soil over the 60-minute period, and the resulting flux calculated would not be a good estimate of soil fluxes in the absence of a sealed chamber. If polynomials are fit to data in for which the slope between 30 and 60 minutes is smaller than the slope between 1 and 15 minutes, again it sounds like you're seeing a chamber artifact where concentrations in the chamber are

approaching equilibrium with concentrations in the soil, and you might be better o9 using data from 1 to 30 minutes.

Response: Using the coefficient of the non-quadratic term in a second order polynomial is equivalent to estimating the flux from the initial slope of the curve

Line 249: Instead of 'scaling up' I think it would be more precise to say that you're estimating a flux that is representative of the entire plot, and it's being done by calculating a weighted mean of fluxes from the basin and interrow chambers. You could write something like "We estimated a flux representative flux for each plot ($N_2O_{plot}$) by calculating a weighted mean of fluxes from the basin and interrow chambers. We used weighting factors of 0.12 for basin . . .".

Response: We thank the reviewer for this constructive suggestion and have rephrased as follows (Ln 249 – 253): *We estimated a representative flux for each plot (area-weighted cumulative $N_2O$ emission) by calculating a weighted mean of fluxes from the basin and interrow positions. Weighing factors of 0.12 and 0.88 were used for basin and interrow areas, respectively, in CA treatments (CA and CA+BC), while a factor of 0.50 was applied to both inrow and interrow areas in conventional treatments (Conventional and ConventMM).*

Line 259: indicate when extractions were conducted relative to sampling (e.g., x hours, the next day, etc)

Response: We have added the following sentence (Ln 259 - 260): *The soil samples were extracted the same day, within 5 hours after sampling.*

Line 280: I would change "dividing the scaled cumulative N2O emissions . . ." to something like "dividing the weighted mean N2O emissions" or "dividing $N_2O_{plot}$ emissions. . ."

Response: Since we earlier mentioned how scaled cumulative N2O emissions were calculated (Ln 249 - 253), we have rephrased the sentence as follows (Ln 281 - 284): *Yield-scaled $N_2O$ emissions (kg $N_2O$-N $kg^{-1}$ grain yield) and N-yield scaled emissions (kg $N_2O$-N $kg^{-1}$ grain N) were estimated for each season by dividing the area-weighted cumulative $N_2O$ emissions with grain yield or N content of the grain (N concentration × grain yield).*

Line 281: I would change "scaling factor" to "weight"
Response: done

Line 283: I would change "scaling factor" to "weight"
Response: done

line 293, you indicate that random effects were introduced to account for repeated measurements. I need more justification for this: typically, one assumes that repeated measures of the same plot will be correlated to some extent. And since

statistics are only conducted on cumulative variables (which is perfectly reasonable, and a good way to avoid issues of autocorrelation in repeated measures), I'm not sure why time is discussed as an effect at all.

Response: Repeated measurements (frequency) was used as a random effect on hourly N2O fluxes and mineral N, not cumulative N2O emissions (see Tables S1 and S2). N2O flux data were used for regression analyses with WFPS and mineral N.

Line 296-298: I find it rather remarkable that no data transformations were required (I've never seen field data like this that didn't).

Response: For cumulative $N_2O$ emissions, we checked the normality of residuals. Visual inspection of the QQ plots showed that residuals were near normal without transformation as shown in fig 1. See also our response to reviewer #1, who raised similar concerns.

[Figure]

Fig 1: Residuals of cumulative $N_2O$ emissions

For N2O fluxes, The QQ plot shows deviations from normality, which is common for $N_2O$ flux data. However, mixed-effects models are robust to violations of normality in residuals (Zuur et al., 2009), especially with large sample sizes. We tested log transformation, but it did not improve model fit (AIC/BIC remained similar), so we retained the original scale for interpretability.

[Figure]

Fig 2: N$_2$O fluxes

[Figure]

Fig 3: Original N$_2$O fluxes vs log transformed N$_2$O fluxes

We have rephrased and included this information in the manuscript as follows (Ln 296 - 301): We validated model assumptions by checking quantile plots of residuals against fitted values. Visual inspection of QQ plots showed that residuals were approximately normally distributed for cumulative N$_2$O data. However, hourly N$_2$O flux data were transformed and did not substantially improve model fit, so we retained the original scale of interpretability. Mixed-effects models are robust to mild non-normality (Zuur et al., 2009).

Line 333: change "were" to "when"

Response: done

Line 349: change "second respectively" to "second season, respectively"

Response: done, thank you.

Figure 2 & 3: I think you want to combine these into a single figure—I don't think it's necessary to compare Conv to convMM in one figure and then compare Conv to the CA treatments in a separate figure, especially since you're presenting post-hoc contrasts.

Response: In this study we tested the effect of (1) rotation and (2) tillage. This is stated in Ln 116 – 119 as follows: *Specifically, we compared crop rotation (pigeon pea – maize) with maize monocropping under conventional tillage (ConventMM). In addition, we compared pigeon pea – maize rotation under three practices, i.e. conventional tillage, CA (reduced tillage), and CA in combination with biochar (CA+BC).*

We therefore think it is relevant to present the data in two separate figures (Fig. 2 and 3). Figure 2 compares the effect of rotation under conventional tillage and figure 3 the effect of different tillage practices and biochar on mineral N.

The higher NH4+ in convMM than all other treatments in season 1 seems unexpected, and (maybe) higher or equivalent to the CA pigeon pea treatments in both years? Something that may be worth discussing.

Response: Thank you for the comment, we now discuss this finding as follows (Ln 489 – 499): *Generally, NH4+ and NO3- contents were more variable in the first season (May – October) than the second season (October – January) (Fig 1b, 1c, 3). At the onset of the experiment, mineral N was most likely from mineralisation of chemically mulched grasses having grown on the fallow for 3 years prior to the experiment. Higher NH4+ in maize than pigeon pea in the first season under conventional tillage might be attributed to differences in crop phenology. Pigeon pea is a slow starter; its nodulation and peak biological $N_2$-fixation typically occur around 80 days after sowing. Therefore, during the early establishment phase, April – June, pigeon pea likely relied on native soil NH4+. Legumes generally show strong affinity for soil $NH_4^+$ in their early growth stages reflecting the lower energy cost for its assimilation. Consequently, efficient soil N uptake by pigeon pea in the early stages reduces soil $NH_4^+$ levels compared to maize, which generally exhibits slower early-season N uptake.*

Line 388: Delete the text "Fig. 4" -- it does not present cumulative emissions

Response: deleted

Line 392: Interpreting the treatment main effect in the presence of a significant treatment x position interaction (which says that the treatment effect depends on the chamber position) is complicated—another reason to use the weighted mean flux for each plot as the response variable. But I do believe interpretation of a significant main e9ect in the presence of an interaction in an RCB design does indicate a main e9ect that is over and above the treatment x block interaction.

Response: We agree, in addition to the position effect, we have now added area-weighted cumulative N2O emissions as a response variable (Table 3).

Figures 5 and 6: I think you want to combine these into a single figure—I don't think it's necessary to compare Conv to convMM in one figure and then compare Conv to the CA treatments in a separate figure, especially since you're presenting post-hoc contrasts.

Response: In this study we tested the effect of (1) rotation and (2) tillage. This is stated in Ln 116 – 119 as follows: Specifically, we compared crop rotation (pigeon pea – maize) with maize monocropping under conventional tillage (ConventMM). In addition, we compared pigeon pea – maize rotation under three practices, i.e. conventional tillage, CA (reduced tillage), and CA in combination with biochar (CA+BC).

We therefore think it is relevant to present the data in two separate figures. Figure 5 compares the effect of rotation under conventional tillage and figure 6 the effect of different tillage practices and biochar on $N_2O$ emissions.

Table 3:

- It would be helpful to have a reminder in the Table caption that the weighted mean fluxes were used to calculate yield-scaled emissions.
- I would add something to the discussion explaining why we would care about N yield scaled emissions.
- Why are the season comparisons (capital letters) only included for ConventMM and not the other treatments?
- I would explain in the table caption why ConventMM is not included in the treatment tests (lower case letters) in Season 1 (presumably because comparing maize yields to pigeon pea yield isn't a useful comparison).

- I think it would be useful to include the P values rather than "ns." It provides the reader with more information and context for interpreting the results. A P value of 0.08 is very different from a P value of 0.5

Response: We have now updated Table 3 caption as follows (Ln 417 – 426): *Grain yield, grain N yield, weighted N2O, yield scaled N2O emissions and N yield scaled N2O emissions during the first and second rain season, in northern Uganda. When*

*calculating yield-scaled N2O emissions, a weighing factor of 0.12 in planting basins and 0.88 in interrows was used in CA treatments (CA and CA+BC), and a weighing factor of 0.50 was used for both inrows and interrows in conventional treatments (ConventMM and Conventional). Means are shown with standard errors of means (N=4). Uppercase letters compare seasons specifically for a monocrop treatment (ConventMM), maize was grown in both seasons. Lowercase letters compare treatments with rotation (Conventional, CA, CA+BC) within a season. Different letters represent significant differences (p < 0.05), determined at 5% level using Tukey test.*

Line 434: Again, and throughout the manuscript, I would report the actual P value rather than P>0.05 or P<0.05

Response: We have now reported the actual p values

Line 434-443: I would just make it explicit here that you are not including ConvMM in treatment comparisons for season 1 since maize was grown in that treatment rather than pigeon pea, as in all the other treatments.

Response: done

Line 442: "significantly higher" needs to be changed: Maize yield was higher, but yield-scaled emissions were lower

Response: Thank you for noticing this. We have corrected the phrase, and it now reads as follows (Ln 432 - 438): *Maize yield in ConventMM were significantly higher in the second season compared to the first season, while yield scaled emissions were greater in the first season*.

Line 461: I think it would be useful to compare emissions to conventional agricultural settings as well—control plots from other experiments in East Africa. It may also be useful to provide emissions—including yield-scaled emissions--from conventional management using fertilizer as context for the conventional and CA plots in this study, while also discussing issues associated with no-input agriculture such as nutrient depletion—how long could these different practices remain sustainable?

Response: We thank the reviewer for this suggestion, and now compare our results with those reported for conventional treatments in East Africa where no fertilizer was applied. The main challenge here is to find adequate literature reporting studies that did not use inorganic fertilizers. We found only two studies which are now

included the discussion (Ln 511 – 514): *Area-weighted cumulative $N_2O$ emission in the conventional treatments ranged from 0.3 to 0.6 kg N ha$^{-1}$ in the 1$^{st}$ and 2$^{nd}$ season, respectively. These results are consistent with Baggs et al., 2006 and Bwana et al., 2021 who recorded $N_2O$ emissions of 0.2 and 0.5 kg $N_2O$-N ha$^{-1}$ in conventional treatments without fertilization.*

To address the issue of no-input agriculture, we now include the following paragraph (Ln 577 – 590): *A key challenge for the sustainability of unfertilized agroecosystems is the management of soil nutrient balances. While biochar amendments in CA systems can effectively reduce $N_2O$ emissions and maintain crop productivity, these systems gradually deplete soil nutrient reserves. Although $N_2O$ is a powerful GHG, it represents a relatively small component of the total annual N budget, often less than 1 kg N ha$^{-1}$. The primary pathway of nutrient removal is the export of grain, which removes a significant amount of N from the system. While biological $N_2$-fixation from pigeon pea can fix up to 110 kg N ha$^{-1}$, some of this input can as well be exported from the field in the stalks, and the remaining N in the form of decaying roots, rhizodeposits and leaf litter is often insufficient to fully compensate for the N removed by the exported grain. Furthermore, the biologically fixed N is prone to rapid mineralization and subsequent loss through leaching or other pathways before the next crop can utilize it. To achieve long term sustainability, an intergrated approach is required, including targeted fertilization to prevent continuous nutrient mining and to ensure the long-term viability of the agroecosystem.*

Line 472: https://doi.org/10.1029/2020JG005742 could be helpful for the rewetting discussion.

Response: We greatly appreciate the provided literature and have rephrased as follows (Ln 465 - 468): *A similar, though smaller emission peak was observed in June when abundant rainfall terminated a dry spell. Rewetting of dry soil triggers $N_2O$ emissions likely due to increased nitrification and denitrification fueled by release of readily available N and C from dead microbial biomass (Namoi et al., 2019), and it is an important source of $N_2O$ emissions in seasonally dry ecosystems (Hickman et al., 2020).*

Line 487: much of the fixed N may be in the harvested biomass, limiting the amount returned to soil.

Response: We have now included this in our discussion as follows:

*A key challenge for the sustainability of unfertilized agroecosystems is the management of soil nutrient balances. While biochar amendments in CA systems can effectively reduce $N_2O$ emissions and maintain crop productivity, these systems gradually deplete soil nutrient reserves. Although $N_2O$ is a powerful GHG, it represents a relatively small component of the total annual N budget, often less*

*than 1 kg N ha$^{-1}$. The primary source of nutrient removal is the export of grain, which removes a significant amount of N from the system. While biological N$_2$-fixation from pigeon pea can fix up to 110 kg N ha$^{-1}$, some of this input can as well be exported from the field in the stalks, and the remaining N in the form of decaying roots, rhizodeposits and leaf fall is often insufficient to fully compensate for the N removed by the exported grain. Furthermore, the biologically fixed N is prone to rapid mineralization and subsequent loss through leaching or other pathways before the next crop can fully utilize it. To achieve long term sustainability, an integrated approach is required, including targeted fertilization to prevent continuous nutrient mining and to ensure the long-term viability of the agroecosystem (Ln 575 – 588).*

Line 494: I don't think NH4 is higher in CA systems than conventional systems—isn't the NH4 in ConvMM no different than the CA treatments? And if ConvMM is higher than conv, it doesn't follow that the mineralization of pigeon pea residues is responsible for the difference. It also conflicts with the statement in line 486.

Response: Thank you for the comment, we have deleted the sentence.

Line 514-516: this sentence is missing something, or 'nutrient cycling' should be deleted.

Response: 'nutrient cycling' was deleted

Line 540-542: You may want to rephrase this. If no treatment effects were expected, it would seem to undermine a justification for the entire experiment (i.e., readers may ask why measurements were conducted in a year when no treatment e9ect was expected, instead of at a time when the treatments would be expected to have an e9ect).

Response: Thank you, we have deleted the statement. See also our response to reviewer #1.

Line 566: delete "We established that"

Response: deleted